# Ketogenic diet suppresses colorectal cancer through the gut microbiome long chain fatty acid stearate

Mina Tsenkova [1], Madita Brauer[1,2], Vitaly Igorevich Pozdeev[1], Marat Kasakin[3], Susheel Bhanu Busi[3,4], Maryse Schmoetten[1], Dean Cheung[1], Marianne Meyers[1], Fabien Rodriguez[1], Anthoula Gaigneaux [1], Eric Koncina [1], Cedric Gilson [1], Lisa Schlicker[3], Diran Herebian[5], Martine Schmitz[1], Laura de Nies[3], Ertan Mayatepek [5], Serge Haan [1], Carine de Beaufort[3,6], Thorsten Cramer [7], Johannes Meiser [8], Carole L. Linster [3], Paul Wilmes [1,3] & Elisabeth Letellier [1] ✉

Colorectal cancer (CRC) patients have been shown to possess an altered gut microbiome. Diet is a well-established modulator of the microbiome, and thus, dietary interventions might have a beneficial effect on CRC. An attenuating effect of the ketogenic diet (KD) on CRC cell growth has been previously observed, however the role of the gut microbiome in driving this effect remains unknown. Here, we describe a reduced colonic tumor burden upon KD consumption in a CRC mouse model with a humanized microbiome. Importantly, we demonstrate a causal relationship through microbiome transplantation into germ-free mice, whereby alterations in the gut microbiota were maintained in the absence of continued selective pressure from the KD. Specifically, we identify a shift toward bacterial species that produce stearic acid in ketogenic conditions, whereas consumers were depleted, resulting in elevated levels of free stearate in the gut lumen. This microbial product demonstrates tumor-suppressing properties by inducing apoptosis in cancer cells and decreasing colonic Th17 immune cell populations. Taken together, the beneficial effects of the KD are mediated through alterations in the gut microbiome, including, among others, increased stearic acid production, which in turn significantly reduces intestinal tumor growth.

Worldwide, colorectal cancer (CRC) is the third most prevalent cancer, with over 1.9 million new cases recorded in 2020[1]. CRC incidence and mortality rates have been associated with various lifestyle-related risk factors, in particular dietary habits. Recent years have seen a rise in CRC incidence in persons aged 50 years and under, further underlining the importance of studying these factors[2]. The ketogenic diet (KD) has been shown to possess anti-cancer properties in the context of CRC, and several studies have focused on the importance of the restriction

[1]Department of Life Sciences and Medicine, Faculty of Science, Technology and Medicine, University of Luxembourg, Esch-sur-Alzette, Luxembourg. [2]Institute for Advanced Studies, University of Luxembourg, Esch-sur-Alzette, Luxembourg. [3]Luxembourg Centre for Systems Biomedicine, University of Luxembourg, Esch-sur-Alzette, Luxembourg. [4]UK Centre for Ecology and Hydrology, Wallingford, United Kingdom. [5]Department of General Pediatrics, Neonatology and Pediatric Cardiology, Medical Faculty and University Hospital Düsseldorf, Heinrich Heine University, Düsseldorf, Germany. [6]Pediatric Clinic, Centre Hospitalier de Luxembourg, Luxembourg, Luxembourg. [7]Department of General, Visceral, Children and Transplantation Surgery, RWTH University Hospital Aachen, Aachen, Germany. [8]Department of Cancer Research (DOCR), Luxembourg Institute of Health, Luxembourg, Luxembourg. ✉e-mail: elisabeth.letellier@uni.lu

of energy intake to slow the growth of rapidly proliferating cells, as well as on anti-cancer signaling through ketone bodies[3]. One recently published study investigated the effects of the KD on CRC development and described a mechanism through which KD exerts its anti-cancer effects, namely through the ketone body β-hydroxybutyrate acting on the surface receptor Hcar2, inducing the transcriptional regulator *Hopx*, thus altering gene expression and inhibiting cell proliferation[4]. These studies often focus on the effects of KD on the cancer cells themselves, and rarely on other cells of the tumor microenvironment, such as immune cells[5,6] and cancer-associated fibroblasts[7], or on external factors, such as the gut microbiome. Specifically in CRC, cancer progression is accompanied by a state of dysbiosis of the gut microbiome[8–10]. The study of the role of the different bacteria in CRC is of particular interest, considering the affected tissue type is in contact with the gut microbiome. Until recently, few studies had investigated the effects of the KD on the gut microbiome in health and disease[11], but this has now changed, with a number of new studies published over the last three years[6,8]. These studies remain largely descriptive regarding dietary-induced gut microbiome changes. However, the role of the gut microbiome in maintaining the reported anti-cancer effects of the KD in the context of CRC has not yet been investigated. Therefore, we set out to investigate the importance of the gut microbiome in the context of KD consumption in CRC. We first generated a humanized gut microbiome mouse model of CRC, through the transplantation of stool samples from five healthy human donors. We then combined this model with a therapeutically administered KD and were able to confirm the cancer-suppressing properties of the KD. Most importantly, we could demonstrate, for the very first time, the causal role of the gut

microbiome in maintaining this effect, through transplantation of the microbial community. We report long-lasting alterations in gut microbiome function in the absence of maintained selective pressure by the KD. In our experiments, fecal free stearic acid levels were increased not only upon KD consumption, but also upon cecal transplantation, suggesting that the shift in the microbial community contributes to the changes in the fatty acid pool. Importantly, stearate-producing members were enriched in ketogenic conditions, whereas consumers were depleted and supplementation of stearic acid reduced tumor burden in vivo.

## Results

### KD leads to reduced colonic tumor burden and immune inflammatory response in CRC

To study the relevance of the gut microbiome in CRC development in a model relevant to human disease, we first established a representative healthy human microbiome by mixing fecal samples from five healthy donors (5HD; Supplementary Fig. 1a for patient metadata). The achieved 5HD mix recapitulated the compositions of the individual donor samples, as evidenced by family, genus and species level analysis of whole genome shotgun sequencing (WGS, Supplementary Fig. 1b–d, respectively), while fecal metabolite levels varied across samples (Supplementary Fig. 1e). We performed fecal microbial transplantation (FMT) of the 5HD mix into germ-free (GF) and antibiotic-treated specific-pathogen-free (SPF) mice in an AOM/DSS CRC model (Fig. 1a, b, details can be found in the Methods section) followed by a switch from standard to ketogenic diet after the last cycle of DSS. We utilized both GF and SPF mice to leverage the fully functional immune system of SPF mice to study immune responses in

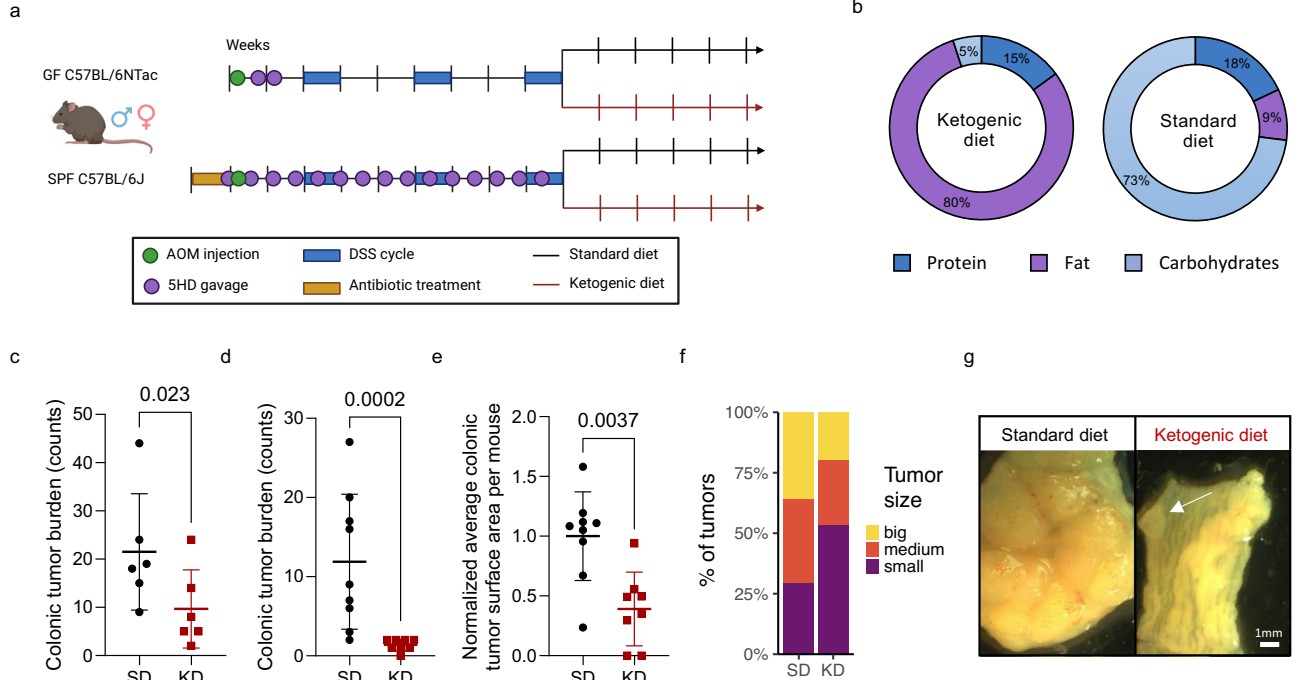

**Fig. 1 | Ketogenic diet leads to lower colonic tumor incidence and smaller tumor size in an inflammatory model of CRC. a** Schematic representation of the dietary AOM/DSS experimental setup in the GF (top) and SPF (bottom) facilities (Created in BioRender. Rodriguez, F. (2024) BioRender.com/w69g496). **b** Compositions of the KD and SD, with % of calories from the indicated source. **c** Colonic tumor burden in GF KD- and SD-fed mice. **d**–**f** Colonic tumor burden (**d**), normalized average colonic tumor surface area per mouse (**e**) and tumor surface area distribution (small=first tertile, medium=second tertile, big=third tertile; one-sided right-tailed Chi-squared test χ2 (2, $N=87$) = 3.4, $p=0.184$; **f**) in SPF KD- and

SD-fed mice. **g** Representative images of colons from one KD- and one SD-fed mouse in the SPF facility. Scale bar = 1 mm. Data is shown as mean ± SD in (**c**), (**d**) and (**e**). Data in (**c**) shows $n=6$ mice per condition from one experiment. Data in (**d**), (**e**) and (**f**) shows $n=8$ and 9 mice in the KD-fed condition and the SD-fed conditions respectively, pooled from two independent experiments in the SPF facility. Analysis of deviance of a negative-binomial generalized linear model regressing the number of tumors against treatment in (**c**), (**d**). Two-sided Mann-Whitney U test in (**e**). Source data are provided as a Source Data file.

CRC and the unique advantage of GF mice allowing engraftment of a human microbiome into mice in the absence of competition with a resident mouse microbiome. Antibiotic-treated SPF mice were subjected to bi-weekly FMT during the induction of CRC to maintain enrichment of the donor fecal microbiome in relation to the residual host mouse microbiome. FMT was ceased at diet change to allow for a natural evolution of the gut microbiome in response to the diet and CRC progression. No significant difference in body weight between the standard diet (SD) and ketogenic diet (KD) groups was observed throughout the experiments, although dips in mouse body weight were observed following the antibiotic treatment and following each cycle of DSS (Supplementary Fig. 2a (GF) and 2b (SPF)). There was no difference in colon length between groups (Supplementary Fig. 2c, GF). KD-fed mice exhibited a lower colonic tumor burden (Fig. 1c (GF) and Fig. 1d (SPF)) and a smaller tumor size (Fig. 1e, f and g (SPF)) than SD-fed mice.

Characterization of full-length lamina propria (LP), mesenteric lymph node (MLN) and spleen immune cell profiles (Supplementary Fig. 2d (GF) and 2e (SPF)) showed a reduction in LP CD4+ T cell populations in KD-fed mice. We particularly observed a decrease in IL17-producing T cells (CD4+ IL17+) (Supplementary Fig. 2f), a pathogenic subpopulation of CD4+ T cells known to play a critical role in CRC development as evidenced by reduced expression of IL-6, STAT3, IFNγ, and TNFα, leading to fewer and smaller tumors in IL-17A-deficient mice[12]. Only LP natural killer T cells and type 3 innate lymphoid cells (GF LP) were increased – both have been associated to gut homeostasis and health[13]. The expression of colonic S100A8, a subunit of calprotectin (a biomarker of inflammation used in inflammatory bowel disease[14]), as well as fecal lipocalin-2 were however unchanged in KD-fed mice, suggesting an overall similar inflammatory milieu of the gut lumen (Supplementary Fig. 2g (GF), h and i (SPF)). Thus, it remains unclear whether the observed reduction in immune cell populations is a result mediated by the KD or if it is a direct consequence of decreased tumor development. Conducting time-course analyzes at various stages of the AOM/DSS-induced carcinogenic process would be essential to elucidate the temporal sequence of events leading to the observed reduction in colonic tumor burden.

To further assess the importance of KD in modulating inflammation in CRC, we employed a low-grade inflammatory model, in which mice received four doses of AOM and were randomized into two subgroups - GF and 5HD mix FMT, each with a KD and SD branch (Supplementary Fig. 3a, b). We did not observe any significant differences in colonic tumor burden between groups (Supplementary Fig. 3c), nor in immune cell profiles (Supplementary Fig. 3d), suggesting that the KD specifically exerted its anti-cancer effect in an inflammatory setting. The differences in immune populations between the KD and SD in the AOM/DSS model, which are absent in this low-grade inflammatory model, may be due to the lack of damage-associated molecular patterns (DAMPs) from DSS-induced epithelial damage. This suggests that the KD attenuates chronic inflammation activated by such damage, potentially leading to a reduced tumor burden in this inflammation-driven CRC model. Of note, long-term KD feeding induced fatty liver and fibrosis in all sub-groups (Supplementary Fig. 3e), as previously reported for long-term KD consumption[15].

### KD induces a distinct host metabolic profile
Ordination analyzes based on the underlying β-diversity among host plasma samples collected from animals undergoing ketogenic dietary intervention in the AOM/DSS inflammatory model of CRC, revealed clustering based on dietary condition (Supplementary Fig. 4a (GF) and Supplementary Fig. 4b (SPF)). β-hydroxybutyrate plasma levels were elevated in KD-fed mice (Supplementary Fig. 4c (GF) and 4d (SPF)), confirming ketosis. KD-fed mice presented altered amino acid, fatty acid, sugar, carbohydrate, vitamin, and bile acid metabolism (Supp. Data File 1 and Supplementary Fig. 4e (GF), Supp. Data File 2 and Supp

Fig. 4f (SPF)). Furthermore, mannose was increased in KD-fed mice (Supp. Data Files 1 (GF) and 2 (SPF)). Interestingly, mannose has been reported to possess antitumor properties[16], impairing tumor cell growth in vitro and in vivo[17], and targeting tumor-associated macrophages[18]. While treatment with β-hydroxybutyrate showed no effect, treatment with the ketone bodies acetone and sodium acetoacetate tended to reduce the differentiation of naive T cells into Th17 cells (Supplementary Fig. 4g and h), which play an important role in the modulation of colorectal tumorigenesis[19] and are affected by KD in a dose-dependent manner[5]. Treatment with a commercially available lipid mixture mimicking KD (containing arachidonic, linoleic, linolenic, myristic, oleic, palmitic and stearic acid) led to a dose-dependent reduction in Th17 differentiation (Supplementary Fig. 4i), suggesting that Th17 cells are susceptible to fatty acid modulation. Collectively, these data indicate that KD induces systemic functional metabolic changes in the host, which are not limited to the induction of ketosis.

### The anti-cancer effects of the KD are mediated by the gut microbiome
The gut microbiome may affect CRC development either by directly interacting with host cells (attachment, invasion and translocation), or indirectly, through bacterial metabolism (genotoxins, SCFAs and others)[20]. We first excluded bacterial translocation as a primary factor in the KD-mediated phenotype, as mucus layer thickness (Supplementary Fig. 5a, b (GF)), LPS levels in plasma (Supplementary Fig. 5c (GF) and 5 d (SPF)), and occludin expression (Supplementary Fig. 5e, f (SPF)) remained unaltered upon KD consumption, in line with Ang et al.[5]. We next performed cecal microbial transplantation (CMT) of samples collected from the SPF diet mice (Fig. 1d) into a next generation of SD-fed GF mice undergoing the same cancer model (Fig. 2a, Supplementary Fig. 6a). We did not observe a statistically significant difference in colonic occludin expression in ketogenic cecal recipient mice (KC, Supplementary Fig. 6b and c) when compared to standard cecal recipients (SC), however we did detect a decrease in S1008A levels (Supplementary Fig. 6d). Strikingly, KC mice exhibited a lower colonic tumor burden than SC mice (Fig. 2b). Average colonic tumor size per mouse did not differ between groups (Fig. 2c), however, KC mice exhibited a smaller proportion of large tumors (Fig. 2d, e). Furthermore, colonic mRNA IL17 levels were decreased in KC mice compared to SC mice (Supplementary Fig. 6e), in line with the decreased number of IL17-producing T cells in the lamina propria of KD-fed mice (Supplementary Fig. 2f). Altogether, our results demonstrate the causal role of the gut microbiome in mediating the previously observed KD-driven anticancer effect.

### KD induces a distinct fecal microbial profile
Ordination analyzes based on the underlying β-diversity among fecal samples collected from KD- and SD-fed AOM/DSS mice revealed clustering of 5HD mix used for FMT with the individual healthy donor stool samples and with the murine stool samples collected three days after FMT (TG, Fig. 3a (GF) and 3b (SPF)), confirming the validity of the 5HD mix used for transplant and its engraftment. Furthermore, we observed increasing differences between KD and SD samples over time (Fig. 3c (GF) and 3d (SPF)). Early time points cluster together (T0, collected at diet change), while later time points diverge (T1-T4). Differential analysis revealed *Phocea* spp., *Faecalitalea* spp. (Fig. 3e (GF), Supp. Data Files 3 (GF) and 4 (SPF)) and *Akkermansia* spp., *Intestimonas* spp., *Lachnoclostridium* spp., *Bilophila* spp. and others (Fig. 3f (SPF)), to be more abundant in KD, while *Barnesiella* spp., *Eisenbergiella* spp., *Ruminococcus 1* and *2* spp. (GF), and *Bifidobacterium* spp., *Turicibacter* spp. (SPF), and others were depleted. Furthermore, the differential abundance of various members of *Barnesiellaceae* spp., *Erysipelotrichaceae* spp., *Lachnospiraceae* spp. and *Ruminococcaceae* spp. suggested that dietary composition affects bacterial members of the same families in different manners.

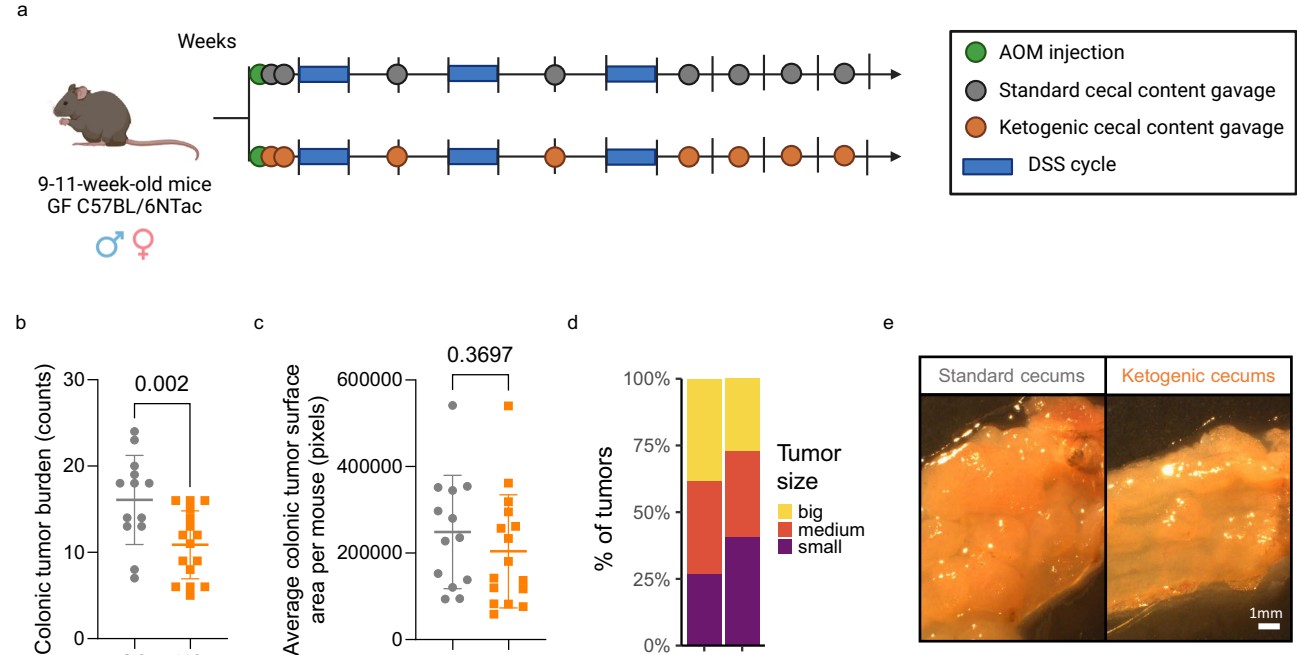

**Fig. 2 | The tumor-inhibitory effect of ketogenic diet is mediated by the gut microbiome. a** Schematic representation of the cecal microbial transfer experimental setup in the GF facility (Created in BioRender. Rodriguez, F. (2024) BioRender.com/w69g496). **b–e** Colonic tumor burden (**b**), average colonic tumor surface area per mouse (in pixels) (**c**) and tumor surface area distribution (small=first tertile, medium=second tertile, big=third tertile; one-sided right-tailed Chi-squared test χ2 (2, N = 384) = 10.3, p = 0.00586; **d** in KC and SC recipient mice.

n = 16 KC recipient mice and n = 13 SC recipient mice. Cecal matter from n = 8 different donor mice per condition was used for CMT. Data shown as mean ± SD in (**b**) and (**c**). Analysis of deviance of a negative-binomial generalized linear model regressing the number of tumors against treatment in (**b**). Two-sided Mann Whitney U test in (**c**). **e** Representative images of the distal portion of the colon from one KC and SC recipient mouse. Scale bar = 1 mm. Source data are provided as a Source Data file.

Investigation of the engrafted gut microbial communities at experimental endpoint revealed distinct KD- and SD-associated microbiomes (Fig. 3g, h), as well as KC- and SC-associated microbiomes (Fig. 3g, h). Although a selective pressure of the KD was observed, gut microbial composition differed between cecal donor and recipient mice, likely due to the absence of continued selective pressure from the KD, or loss of more sensitive and hard-to-engraft bacteria (Fig. 3g). Metagenome-assembled genomes (MAGs), representing abundant members of the microbial community, allowed us to achieve species-level resolution. Noteworthy, some MAGs annotated as CRC-associated bacteria, such as *Rikenellaceae* spp. (light blue) and *Muribaculaceae* spp. (dark blue, Fig. 3i), were detected in standard but not in ketogenic conditions. Interestingly, various *Alistipes* spp. were differentially abundant in all experiments. However, the genus *Alistipes* was only recently described, and although the genus has since then been linked to various diseases, including colitis and CRC, the role of *Alistipes* in disease is still unclear[21]. Two MAGs (*Alistipes* and *COE* species) were only identified in KC, while 12 MAGs were only identified in SC, among which *CAG-873*, *CAG-48* and *Parabacteroides distasonis* (Fig. 3i), also identified in the donor SPF dietary experiment.

**Distinct fecal microbial functional profiles are maintained upon FMT**

As microbiome-mediated effects are based on the functional and metabolic capacities of different microbes, we investigated the fecal metabolic profiles of KD- versus SD-fed and KC versus SC mice, considering these profiles as a direct indicator of changes in the functional capacity of the different microbial communities. Analysis of fecal metabolites revealed distinct clustering of samples by condition in both the SPF dietary and CMT experiments (Fig. 4a, b, respectively). Mice in the ketogenic conditions presented altered amino acid, fatty acid, carbohydrate and SCFA profiles (Supp. Data Files 5 and 6), albeit

to a lesser extent in the CMT experiment. A single detected long-chain fatty acid (LCFA), namely free stearic acid (C18:0, also known as octadecanoic acid or stearate), was elevated in KD (Fig. 4c and Supplementary Fig. 7a for absolute quantities) and KC (Fig. 4d) mice. Strikingly, free stearic acid levels correlated with colonic tumor burden in the SPF dietary experiment (Fig. 4e), with a similar trend observed in the CMT experiment (Fig. 4e), suggesting a functional relevance of LCFA metabolism in CRC.

**KD induces a shift toward stearic acid production, which is maintained upon FMT**

To assess whether and how the microbiome contributes to shaping the LCFA pool, specifically stearic acid, we focused on microbial LCFA metabolism. Most bacteria can produce LCFAs, which are essential bacterial membrane components and signaling molecules. LCFAs are produced in a multistep process by enzymes that are highly conserved among bacteria (Fig. 5a)[22]. Nevertheless, membrane LCFA profiles differ between microbial taxa, varying for instance in chain length and degree of saturation[23]. To investigate how the shift in the microbial communities reported in Fig. 3 affected the intestinal LCFA pool in KD-fed and KC mice, we first mapped our metagenomic sequences against the Kyoto Encyclopedia of Genes and Genomes (KEGG) database to obtain a list of contigs carrying LCFA metabolism-associated KEGG orthologs. These contigs were further annotated at the taxonomic level using Kraken2 to estimate the contribution of different microbial classes to LCFA metabolism in the different dietary conditions. Consistent with the taxonomic shift reported in Fig. 3i, the relative contribution of bacteria from the class Clostridia was higher in the KD groups, while members of the class Bacteroides were diminished resulting in less LCFA metabolism-associated contigs assigned to this class (Supplementary Fig. 7b). Specifically, contigs carrying the microbial acyl-CoA thioesterases TesA and FatA (Fig. 5b)−enzymes

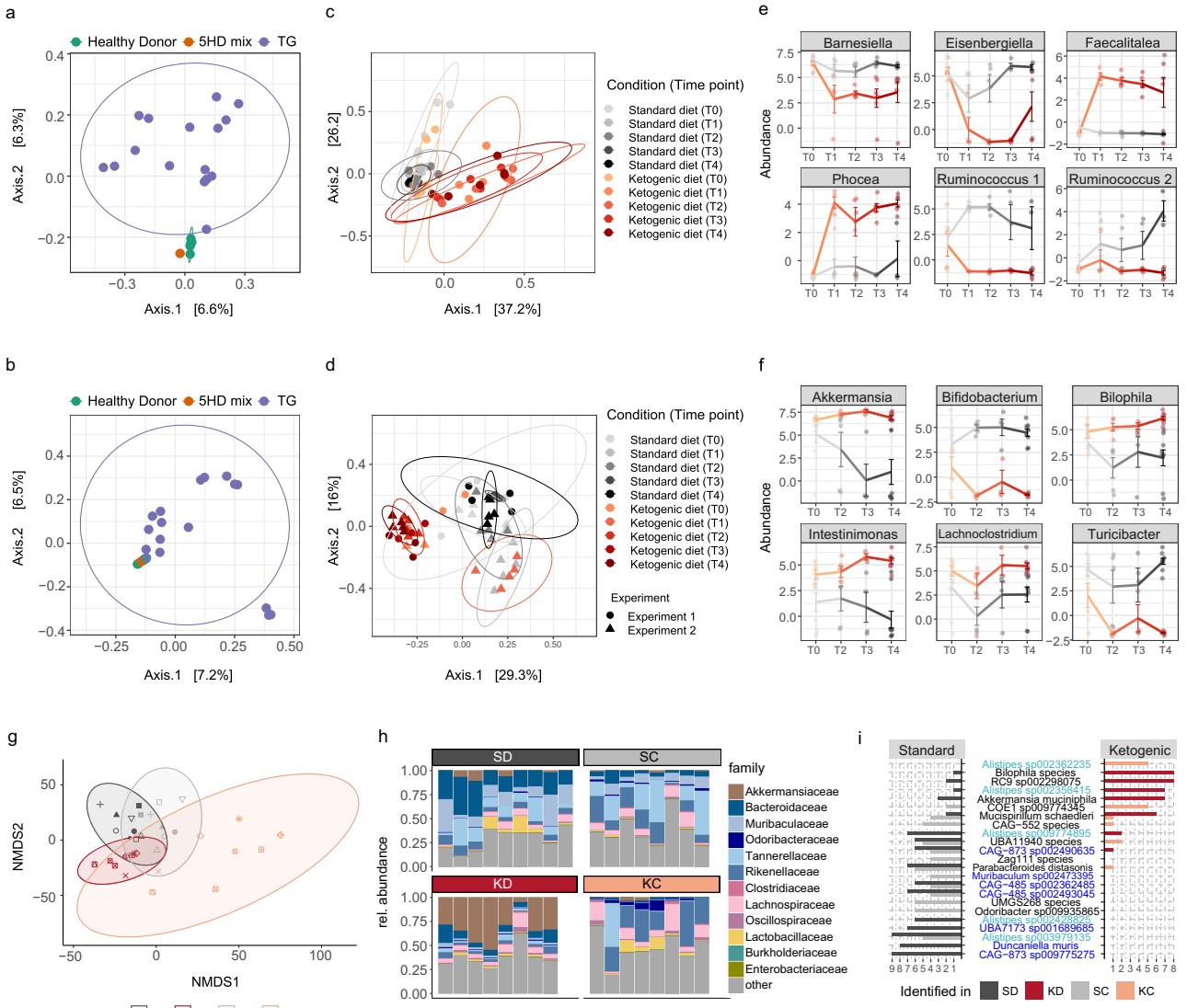

**Fig. 3 | Ketogenic diet alters gut microbiome composition. a–f** β-diversity as assessed by PCoA (% confidence interval = 95) of Bray-Curtis distance in five individual healthy donor stool samples, in a mix of the five healthy donor stool samples (5HD mix) and in murine fecal samples three days after gavage (**TG**, **a** and **b**); β-diversity as assessed by PCoA (% confidence interval = 95) of Bray-Curtis distance in murine fecal samples collected over time after diet change (**c** and **d**); Abundance of selected genera which are differentially abundant at at least one time point (using a generalized linear model on CLR, **e** and **f** data is shown as mean ± SEM) in fecal samples collected over time after diet change from SD- and KD-fed mice in the GF (**a**, **c** and **e**) and SPF (**b**, **d**, and **f**) dietary experiments (T0 at diet change, T1-T4 weeks after diet change), as analyzed by 16S rRNA sequencing. **g–i** β-diversity as assessed by NMDS (% confidence interval = 95) based on an Aitchison distance matrix (**g**),

relative abundance of bacterial families based on Kraken2 IMP output (**h**), and highly characteristic bacterial MAGs (detected in at least six mice in one condition but not more than three mice in the other (diet experiment); detected in at least four mice in one condition but not more than three mice in the other (CMT experiment)), detected in KD, SD, KC and SC murine fecal samples at endpoint, as analyzed by WGS (**i**). In (**i**) *Rikenellaceae* are highlighted in light blue and *Muribaculaceae* in dark blue. Data shows n = 1 5HD mix, n = 5 individual human donor samples in (**a**) and (**b**), n = 15 humanized mouse samples in (**a**) and n = 17 humanized mouse samples in (**b**), n = 6 mice per condition in (**c**) and (**e**), n = 8 SPF KD-fed mice and n = 9 SPF SD-fed mice, pooled from two independent experiments in (**b**), (**d**), (**f**), (**g**), (**h**) and (**i**), and n = 8 KC and n = 8 SC mice from one experiment in (**g**), (**h**) and (**i**). Source data are provided as a Source Data file.

required for the release of free LCFAs, such as stearic acid, from the acyl-carrier[24]—were more frequently assigned to species from the class Clostridia in the KD groups compared to the SD groups. This suggests that in the KD groups, members of the class Clostridia may have contributed more significantly to the intestinal free LCFA pool (Supplementary Fig. 7b). To investigate whether stearic acid is produced by Clostridia, we analyzed a dataset published by Han et al.[25], who recently profiled the metabolome of bacterial culture supernatants. Visualization of log2 fold changes in stearic acid levels in these supernatants revealed that *Bacteroidaceae* and *Tannerellaceae*—two families within the class Bacteroidia—predominantly consume stearic acid in vitro. In contrast, families form the class Clostridia, such as *Lachnospiraceae*

and *Ruminococcaceae*, release stearic acid into the culture supernatants (Fig. 5c). Notably, these families, although not necessarily predominant in the communities, were consistently depleted or enriched across experiments in KD and KC mice, respectively (Fig. 5d). Furthermore, the analysis of two other abundant LCFAs, palmitic and oleic acid, showed that palmitic acid was consumed by most bacteria, with the exception of some *Clostridiaceae*, while oleic acid was released by all bacteria. This finding suggest that stearic acid is a family-specific LCFA among the three analyzed LCFAs (Supplementary Fig. 7c). To further confirm that the source of elevated free stearic acid in the mouse fecal samples was not coming from pre-existing differences in stearic acid levels in the diets themselves, we profiled LCFA

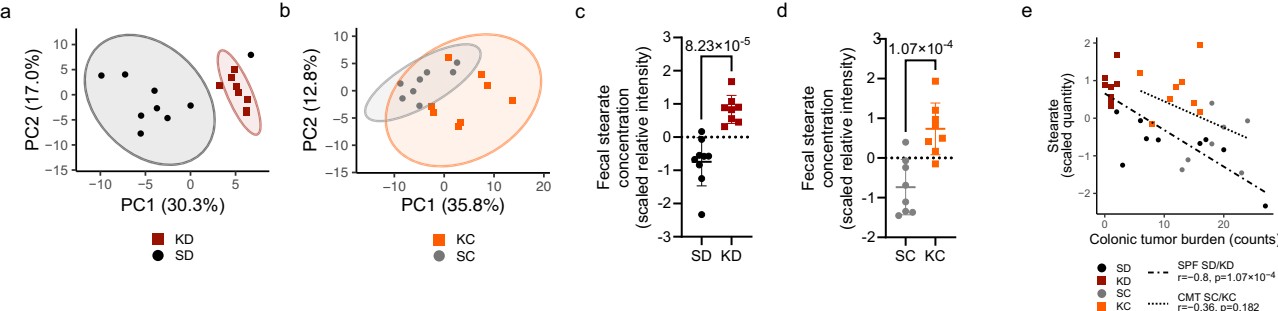

**Fig. 4 | Ketogenic diet alters metabolic profiles, leading to elevated levels of fecal stearate. a, b** PCA (% confidence interval = 95) of identified metabolites from fecal samples from SPF KD- and SD-fed mice (**a**) and from KC and SC recipient mice (**b**), as analyzed by united GC-MS and LC-MS. **c, d** Scaled free stearic acid levels detected in fecal samples from SPF KD- and SD-fed mice (**c**) and from KC and SC recipient mice (**d**) by LC-MS (polar phase). **e** Correlation of colonic tumor burden with free fecal stearic acid levels in SPF KD- and SD-fed mice (dashed line) and in KC and SC recipient mice (dotted line). Two-sided Pearson correlation testing was used. Data shown in (**a**), (**c**), and (**e**, dashed line) are from $n = 9$ SPF SD-fed and $n = 8$ SPF KD-fed mice, pooled from two independent experiments. Data shown in (**b**), (**d**) and (**e**, dotted line) are from $n = 8$ mice per condition. Data shown as mean ± SD, two-sided Mann-Whitney U test in (**c**) and (**d**). Source data are provided as a Source Data file.

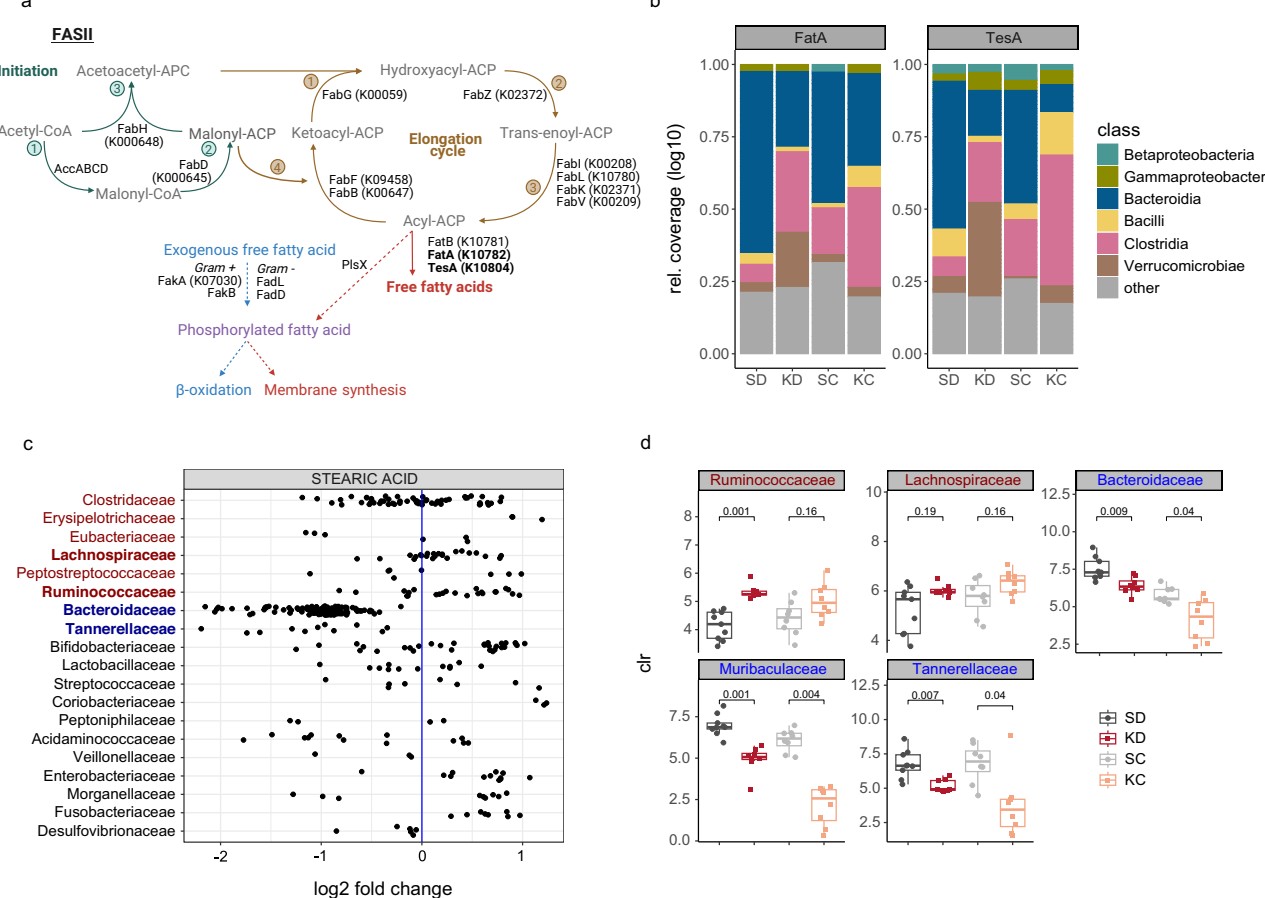

**Fig. 5 | Stearate is a microbial metabolite. a** Schematic representation of type II bacterial fatty acid synthesis created in BioRender. Rodriguez, F. (2024) BioRender.com/w69g496. Core enzymes are shown with their corresponding KEGG orthologs (KOs). Abbreviations: acetyl-CoA carboxylase enzyme complex (AccABCD), β-ketoacyl-ACP synthase 1 (FabB), malonyl-CoA:ACP transacylase (FabD), β-ketoacyl-ACP synthase II (FabF), β-ketoacyl-ACP reductase (FabG), β-ketoacyl-ACP synthase III (FabH), enoyl-ACP reductase enzyme isoforms (FabI, FabK, FabL and FabV), β-hydroxyacyl-ACP dehydratase (FabZ), long-chain-fatty-acid-CoA ligase (FadD), long-chain fatty acid transport protein (FadL), fatty acid kinase (FakA, FakB), acyl-ACP thioesterase (FatA), iron transport protein (FatB), thioesterase 1/protease 1/lysophospholipase L1 (TesA), phosphate acyltransferase (PlsX), Coenzyme A (CoA), acyl carrier protein (ACP). Thioesterases of interest are bolded. **b** Contigs from metagenomic sequencing data carrying the KOs of the bacterial acyl-CoA thioesterases FatA and TesA were assigned to bacterial classes using Kraken2. Relative proportions (log10-transformed coverage of contigs) of relevant bacterial classes are displayed. Remaining bacterial classes were summarized as other. **c** Levels of stearic acid measured in bacterial cultures (in relation to sterile growth medium, shown as log2) of the Sonnenburg dataset published by Han et al. Families of the class Clostridia are listed in red, while families of the class Bacteroidia are listed in blue. **d** Relative abundance (centered-log transformed total read counts) of stearate-producing and -consuming families of interest (bolded in **c**) detected in murine fecal samples from KD-, SD-fed, KC and SC recipient mice at endpoint. Boxplots show medians with 1st and 3rd quantiles. The whiskers from the hinges to the smallest/largest values represent 1.5*inter-quartile range (IQR). Two-sided Wilcoxon signed-rank test with Bonferroni correction. Data shown in (**d**) are from $n = 9$ SPF SD-fed and $n = 8$ SPF KD-fed mice, pooled from two independent experiments (red and black) and from $n = 8$ mice per SC/KC condition (gray and orange). Source data are provided as a Source Data file.

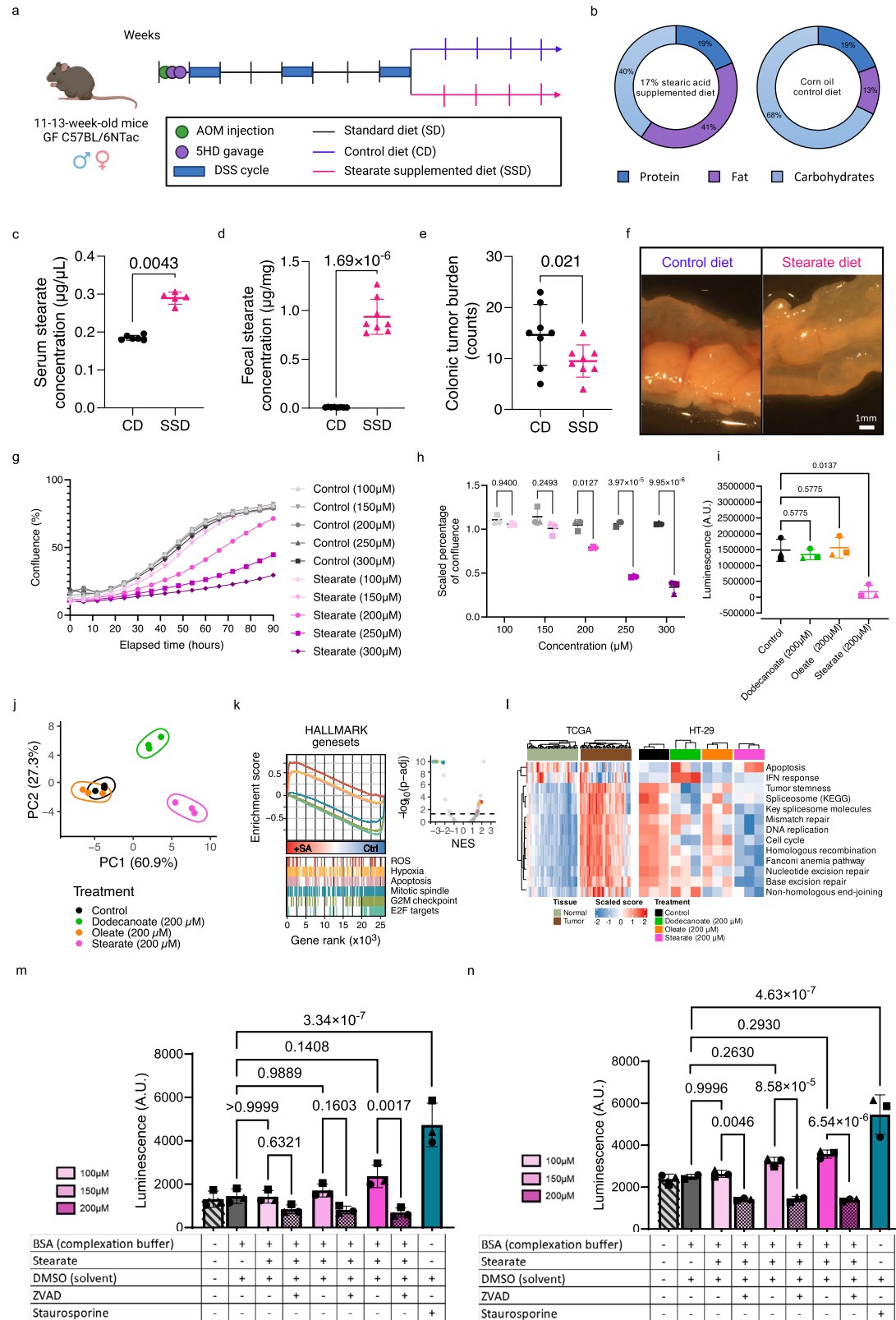

content in the SD and KD and confirmed similar levels of free stearic acid (Supplementary Fig. 7d).

## Stearic acid reduces colonic tumor burden by inducing apoptosis in cancer cells

To further investigate the role of stearic acid in CRC development, we supplemented stearic acid through the dietary regimen to mice

undergoing the same AOM/DSS CRC model described above (Fig. 6a, b and Supplementary Fig. 8a) and confirmed increased free stearic acid levels in the stearate-supplemented diet compared to the control diet (Supplementary Fig. 8b). Mice on stearate-supplemented diet (SSD) also exhibited elevated stearic acid levels in serum and fecal samples (Fig. 6c and d, Supp. Data Files 7 and 8), similar to the levels observed in the fecal samples of mice on KD (Supplementary Fig. 7a). We observed

**Fig. 6 | The microbial metabolite stearate exhibits anti-cancer effects.**
**a** Schematic representation of the dietary stearic acid supplementation experimental setup (Created in BioRender. Rodriguez, F. (2024) BioRender.com/w69g496). **b** Compositions of the control (CD) and stearate-supplemented (SSD) diets, with % of calories from the indicated source. **c, d** Stearic acid concentrations in serum ($\mu g/\mu L$, **c**) and feces ($\mu g/mg$, **d**) from CD- and SSD-fed mice, as detected by LC-MS-based long chain fatty acid quantification. **e** Colonic tumor burden in CD and SSD mice. **f** Representative images of colons from one CD and SSD mouse. Scale bar = 1 mm. Data shows mean ± SD from $n = 6$ CD and $n = 5$ SSD mice in (**c**), from $n = 8$ CD and $n = 8$ SSD mice in (**d**). Two-sided Mann-Whitney U test in (**c**) and (**d**). Data shows mean ± SD from $n = 8$ CD and $n = 8$ SSD mice in (**e**). Analysis of deviance of a negative-binomial generalized linear model. **g, h** HT29 proliferation over time (**g**) or scaled confluence at 72 hours (**h**) after treatment with stearic acid or corresponding control. Data shows mean (**g**) or mean ± SD (**h**) from $n = 3$ independent experiments, indicated by different datapoint shapes (**h**), each with eight

technical replicates per condition. **i** ATP measurement (CellTiterGlo) of HT-29 cells at 72 hours after fatty acid or control treatment ($200\,\mu M$). Data shows mean ± SD of $n = 3$ independent experiments (indicated by different datapoint shapes), each with eight technical replicates. Two-way ANOVA with multiple comparisons in (**h**), (**i**). **j** Principal component analysis of fatty-acid-treated HT-29 cells ($200\,\mu M$). **k** Hallmark gene sets in stearic acid-treated versus control cells. **l** Cell cycle and apoptosis related gene expression in control and fatty-acid-treated HT-29 cells and in tumor and matching normal tissue samples. **m, n** Caspase-3 activity in HT-29 (**m**) and HCT-116 (**n**) cells treated with stearic acid ± broad caspase inhibitor (zVad). Staurosporine 2uM was used as a positive control. Data shown in (**j**), (**k**), (**l**) is from $n = 3$ independent experiments. Data shows mean ± SD of $n = 3$ biological replicates (indicated by different datapoint shapes), in (**m**) and (**n**). Ordinary one-way ANOVA in (**m**) and (**n**), data passed Shapiro-Wilk normality test. Source data are provided as a Source Data file.

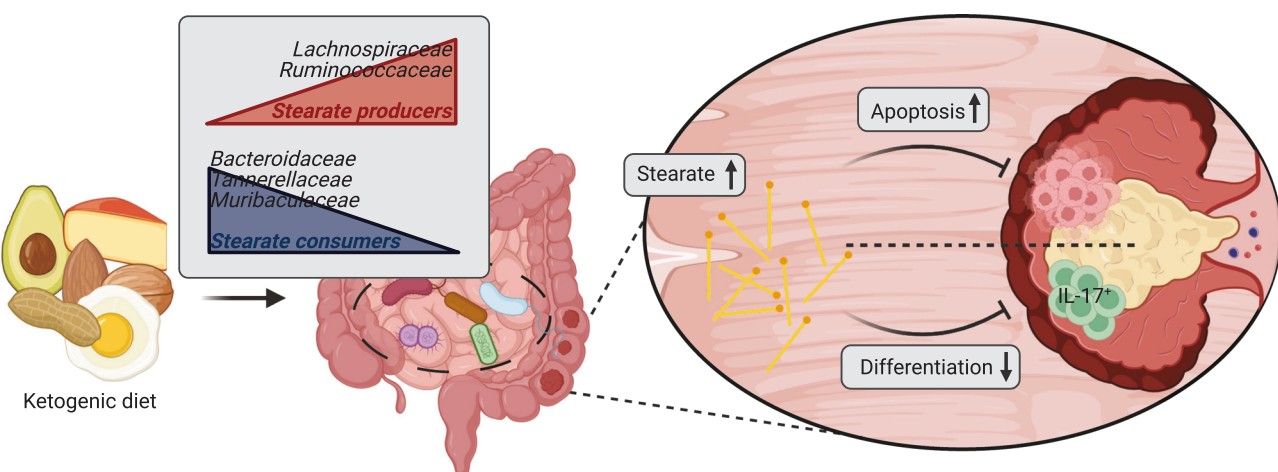

**Fig. 7 | Functional ketogenic-diet-induced changes in the gut microbiome are responsible for its anti-cancer properties.** Ketogenic diet consumption leads to lasting functional changes in the gut microbiome, which are sufficient to reduce colonic tumor burden and which impact the luminal long-chain fatty acid pool. The enriched microbial product stearate demonstrates tumor-suppressing properties by inducing apoptosis in cancer cells and decreasing colonic Th17 immune cell populations. Created in BioRender. Rodriguez, F. (2024) BioRender.com/w69g496.

a reduction in colonic tumor burden (Fig. 6e, f), indicating that stearic acid exerts an anti-tumorigenic effect in the development of CRC. We did not observe any difference in fecal lipocalin-2 levels between groups (Supplementary Fig. 8c), however, we further linked the reduced tumor burden to decreased Th17 cells in vivo (Supplementary Fig. 8d). Furthermore, subsequent T cell differentiation assays demonstrated that stearic acid supplementation specifically inhibited differentiation of CD4⁺ T cells into Th17 cells, a response not observed with other lipids (Supplementary Fig. 8e, f).

Treatment of various CRC cell lines with increasing concentrations of stearic acid led to a dose-dependent reduction in cell growth (Fig. 6g, h, Supplementary Fig. 8g, h, i, j), highlighting stearic acid as an anti-cancer metabolite which exerts direct effects on tumor cells. Furthermore, this observation was specific to stearic acid, as treatment with other LCFAs did not exhibit anti-cancer properties (Fig. 6i). We then performed RNA sequencing on HT-29 CRC cells treated with stearic, oleic and dodecanoic acids to investigate the molecular mechanisms underlying the observed anti-cancer effect. Gene set enrichment analysis (GSEA) suggested an arrest of cell proliferation and the induction of apoptosis upon stearic acid treatment (Fig. 6k, l). Furthermore, stearic acid treatment was able to revert cell cycle and apoptosis related gene expressions observed in control (vector-treated) cancer cells, which are in line with those observed in patient tumor samples compared to matching normal tissue samples in the TCGA dataset (Fig. 6l, Supplementary Fig. 8k). The use of a broad caspase inhibitor was able to abrogate the effects of stearic acid treatment,

confirming the underlying mechanism of stearic acid-induced apoptosis of cancer cells (Fig. 6m, n). In summary, our results show that ketogenic diet consumption leads to lasting functional changes in the gut microbiome, which are sufficient to reduce colonic tumor burden and which impact the luminal long-chain fatty acid pool. The enriched microbial product stearate demonstrates tumor-suppressing properties by inducing apoptosis in cancer cells and decreasing colonic Th17 immune cell populations (Fig. 7).

## Discussion

The importance of the composition and function of the intestinal microbiome can be observed in different disease contexts. Interestingly, recent studies have emphasized that the therapeutic response in cancer is linked to specific bacterial functions, rather than individual bacterial species, suggesting that microbial-derived metabolites play a critical role in driving the effectiveness of therapy[26–28]. In this study we first identified the gut microbiome as a driving factor in the anti-cancer effect of KD in CRC. We next demonstrated that KD alters gut microbial composition and metabolism. Upon KD consumption and following microbial transplantation, we observed an increase in fecal free stearic acid, an LCFA, which showed direct cancer-suppressing effects through the induction of apoptosis in cancer cells, as well as indirect cancer-suppressing effects through suppression of pathogenic colonic Th17 cells (Fig. 7).

The gut microbiome plays a key role in mediating the anti-cancer effects of the KD in the context of a murine model of

inflammation-driven CRC. However, the variability in the response observed (Fig. 2b) suggests that the initial microbiome profile may determine whether an individual is a "responder" or "non-responder" to KD. This concept, explored in epilepsy patients undergoing KD therapy for the reduction of seizures[29] and in cancer treatments like immune checkpoint inhibition, highlights the importance of the gut microbiome. Future studies should focus on identifying biomarkers to predict response and strategies to convert "non-responders" into "responders," paving the way for personalized treatments.

The depletion of dietary carbohydrates is known to provide a growth advantage for certain bacteria, such as *Alistipes* spp., *Bilophila* spp. and mucin-degraders like *Akkermanisa* spp., whereas *Bifidobacterium* spp. and some *Bacteroides* spp. are typically depleted[5,30], most likely due to their preference for dietary carbohydrates over host glycans[31–33]. However, differences observed in experiments investigating carbohydrate restriction and KD as intervention strategy suggest that limiting carbohydrates is not the sole factor responsible for the beneficial effects of the KD. In our study, the analysis of the fecal metabolomes in the KD dietary intervention and CMT experiments identified differences in the LCFA pool, which are directly linked to different microbiome composition and metabolic activity. In addition to their essential role in membrane biogenesis, LCFAs produced by bacteria have signaling functions and play an essential role in host-microbe interactions[34]. The LCFA profile is highly species-, or even strain-, and substrate-dependent and various microbial-derived LCFAs have different effects linking bacterial LCFA metabolism to host health and disease[23,34]. Amongst others, microbiome-derived LCFAs have an immunomodulatory potential[34,35]. We specifically identified stearic acid, an LCFA, in KD and KC mice, which can originate from the diet, the host[36] and the gut microbiome[37]. We observed the stearic acid to be specifically increased in the fecal samples of the KD-fed mice and not in the circulatory system, suggesting that it is not a systemic effect of the KD but rather that it is the KD-driven changes in the gut microbiome that led to differences in stearic acid.

Stearic acid is normally desaturated by the host enzyme SCD1[38], which is downregulated upon reduced glucose intake and has been linked to inflammation[39–41]. Recently, Kindt et al. showed that microbiome-derived acetate activated SCD1 and promoted LCFA desaturation[42]. Furthermore, certain members of the gut microbiome can directly affect stearic acid levels by producing it de novo or transforming it from other LCFAs, such as oleic acid[37], or by acquiring it as an exogenous LCFA for downstream use in membrane lipid synthesis, or LCFA degradation[22,43]. In our experiments, as free fecal stearic acid levels were increased not only upon KD consumption, but also in KC, it is likely that the shift in the microbial community is contributing to the changes in the LCFA pool. Importantly, potential stearate-producing members were enriched by the KD, whereas consumers were depleted (Fig. 5b–d), leading to increased free stearic acid levels in the gut lumen.

Finally, we investigated the effects of stearic acid in our humanized CRC mouse model. Through a stearic acid-enriched dietary intervention, we demonstrated that stearic acid effectively inhibits tumor growth both directly, by inducing apoptosis in cancer cells, and indirectly, by reducing the recruitment of pro-tumorigenic Th17 cells. In previous studies, stearic acid inhibited colony formation in different human cancer cell lines[44], reduced tumor burden in a rat model of mammary carcinoma[45], possibly by inducing apoptosis via signaling through protein kinase C[46], and suppressed CRC in synergy with 5-FU treatment[47].

Taken together, our study highlights the significant role of the gut microbiome and its derived metabolites, in particular stearic acid, in mediating the KD's anti-tumorigenic effects. Most importantly, our findings suggest that oral supplementation with a single metabolite could enhance existing treatment strategies for CRC.

## Methods

### Human sample collection
Human samples were obtained and handled in accordance with institutional guidelines. All human samples used in this project were donated freely, upon written informed consent. Ethical approval was given by the Comité National d'Ethique de Recherche of Luxembourg (MuSt authorization number 201110/05) and the National Commission for Data Protection in Luxembourg. Samples were collected at the donors' homes in accordance with the standard practices established by the Integrated Biobank of Luxembourg (IBBL), in feces collection tubes, then aliquoted and stored at −80 °C until further use.

### In vivo mouse experiments
Animal experiments were performed according to all applicable laws and regulations, with approval from the Animal Experimentation Ethics Committee (AEEC) and the veterinary service of the Ministry of Agriculture, Viticulture and Rural Development of Luxembourg (LUPA2019-13). They ensure that the care and the use of the animals for research purposes is conducted according to the European Union (EU) Directive 2010/63/EU and to the Grand-Ducal Regulation of the 11th of January 2013, regarding the protection of animals used for experimentation. This includes justification for the use of the animals, guidelines for their care and welfare, and the incorporation of the 3Rs (replacement, reduction and refinement). All mouse experiments were carried out at the animal facility of the Luxembourg Center for Systems Biology (LCSB). Ethical guidelines for humane animal experimentation as set forth by the AEEC and the Ministry have not been exceeded.

C57BL/6 J mice were purchased from Charles River France. C57BL/6NTac mice were bred in house. C57BL/6 J mice were housed in AllenTown NexGen Mouse 500 (19.4 cm x 13.0 cm x 38.1 cm) cages with JRS Rehofix Corncob bedding, while C57BL/6NTac mice were housed in AllenTown Sentry Sealed Positive Pressure Individually Ventilated (SPP IVC) cages (13.3 cm × 12.8 cm × 32.5 cm). Food and water were provided *ad libitum*. Animals were maintained on standard rodent chow (A04, SAFE Lab, irradiated at 40kGy) and ketogenic diet (S0382-E210, ssniff Spezialdiäten, irradiated at 25kGy) or on control diet (MD.230304, Envigo, irradiated at 25kGy) and stearate-supplemented diet (MD.230306, Envigo, irradiated at 25kGy). The animals were maintained under standard habitat conditions (humidity: 40–70%, temperature: 22 °C) with a 12:12 light cycle.

### Stool preparation for fecal/cecal microbial transplant
Single human stool samples and single mouse cecal contents were suspended in a 20% glycerol PBS solution at 10 mg of stool per mL. For the mix of five healthy donor stools (5HD), equal volumes of each single suspension were mixed.

### Inflammatory mouse model of CRC
GF and SPF mice were administered a single intraperitoneal injection of sterile-filtered AOM (Sigma-Aldrich, 10 mg/kg mouse body weight). SPF mice were administered a sterile-filtered antibiotic mix (vancomycin hydrochloride (Sigma-Aldrich, 500 mg/L), ampicillin (Sigma-Aldrich, 1 g/L), neomycin (Sigma-Aldrich, 1 g/L) and metronidazole (Sigma-Aldrich, 1 g/L)) via their drinking water for one week prior to AOM injection. Microbiomes were replaced with donor microbiomes via oral gavage. Mice were administered a total of three cycles of DSS, with a two-week-long recovery period between each cycle. DSS was refreshed mid-cycle. Half of the mice were randomly allocated to each dietary condition at the end of the last cycle of DSS. Further details regarding the experimental setups can be found in Supp. Data File 9.

### Long-term lowly-inflammatory mouse model of CRC
GF mice were administered four intraperitoneal injections of sterile-filtered AOM (Sigma-Aldrich, 15 mg/kg mouse body weight), once per

week for four weeks and received donor microbiomes via oral gavage. Half of the mice were randomly allocated to the ketogenic diet one week after the first injection of AOM. Further details regarding this experimental setup can be found in Supp. Data File 9.

## Mouse euthanasia

All mice were injected intraperitoneally with ketamine (150 mg/kg, Nimatek®, Medikamio) and medetomidine (1 mg/kg, Dorbene®, Zoetis) at endpoint and blood was collected from the portal vein followed by cervical dislocation.

## Colonic tumor burden and tumor size assessment

Colons were resected at endpoint and small portions of the proximal colon were snap-frozen for further downstream analyzes. The remaining distal portion of the colon was used for the assessment of tumor burden. Colonic tumors were counted under a Leica Wild M10 microscope and photographed using a Leica MC170HD camera. Colonic tumor surface size was assessed from the photographs using FIJI Image J.

## Lamina propria immune cell isolation

Colons (opened longitudinally) were cut into 2 cm-long pieces and pre-digested in strip medium (RPMI GlutaMAX® (Gibco), 3%FBS (Lonza), 1% P/S (Gibco), 5 mM EDTA (Sigma-Aldrich), 0.154 mg/mL DTT (Merck)) in a bacterial shaker at 800 rpm, 37 °C for 20 minutes. After incubation, the strip medium was collected, filtered through a 70 µM and retained to further immunophenotyping, constituting the intra-epithelial lymphocyte (IEL) fraction. The remaining intestinal pieces were digested (RPMI GlutaMAX® 1%P/S, 5 mg/mL Liberase (Merck Life Sciences), 25 mg/mL DNase (Sigma-Aldrich)) for 30 minutes at 800 rpm, 37 °C in a bacterial shaker. The digest was pressed though a 70 µM cell strainer, washed with FCS-containing RPMI medium and retained for further immunophenotyping. Alternatively, colons were processed according to the Mouse Lamina Propria Dissociation Kit (Miltenyi) instructions, with the following modifications: 1) a bacterial shaker was used (800 rpm, 37 °C) instead of a MACSmix tube rotator. 2) 70 µM cell strainers were used instead of 100 µm cell strainers.

## Spleen and mesenteric lymph node immune cell isolation

Spleens were injected with 1 mL of RPMI GlutaMAX® 1%P/S, 5 mg/mL Liberase (Merck Life Sciences), 25 mg/mL DNase (Sigma-Aldrich), cut up into small pieces (<1 mm$^2$) and incubated at 37 °C for 30 minutes. After incubation, spleens were pressed though a 70 µM cell strainer, washed with FCS-containing RPMI medium and retained for further immunophenotyping. MLNs were pressed through a 70 µM cell strainer without prior enzymatic digestion.

## Cell counting

The cellular concentration of all cell suspensions was quantified using the Cedex XS Analyzer (Roche), by mixing 10 µL of cells with 10 µL of Trypan Blue solution (Roche) and loading 10 µL into the Cedex Smart Slide (ibidi) channel; or using the CASY Cell Counter and Analyzer (OMNI Life Sciences), by diluting 10 µL of cell suspension in 10 mL of CASYton (OMNI Life Sciences) solution.

## Immune cell restimulation

Cell suspensions obtained from colons, spleens and MLNs were distributed in a 96-well plate and either directly stained or restimulated prior to intracellular cytokine staining using 50 ng/mL PMA (Sigma-Aldrich) and 1.5 µg/mL ionomycin (Alfa Aesar) in RPMI GlutaMAX® 10% FBS 1% P/S for four hours at 37 °C. Brefeldin A (eBioscience (Invitrogen)) was added after the first hour.

## Immune cell staining and acquisition (FACS)

Cells were stained with fluorescent extracellular antibody mixes (LIVE/DEAD Fixable Near-Infrared Dead Cell Stain Kit (ThermoFisher Scientific) 1:1000, antibodies 1:100 in FACS buffer) and incubated for 30 minutes at 4 °C or overnight at 4 °C. For the staining of macrophages and dendritic cells, an anti-CD16/32 (1:50) antibody was used to block unspecific binding prior to staining with fluorescent antibodies. After incubation, cells were washed, then fixed and permeabilized using the Cytofix/Cytoperm Kit (BD Biosciences), according to the kit manual. After permeabilization, cells were stained with fluorescent intracellular antibody mixes (antibodies 1:100 in Perm Buffer (from Cytofix/Cytoperm Kit)) and incubated for 30 minutes at 4 °C. After incubation, cells were washed and analyzed in a FACS Canto II or on a FACS LSRFortessa Cell Analyzer. All antibodies are listed in Supp. Data File 10. Data analysis was performed using FlowJo and R was used for graphical representation. All gating strategies for immune cell profiling presented in this article can be found in Supp. Figs. 9 and 10.

## Tissue staining

Liver lobes and portions of colons were included in Tissue Freezing Medium (Leica) and snap-frozen in liquid nitrogen immediately upon collection. Slides were thawed and air dried for 30 minutes a room temperature prior to staining. For Alcian blue staining, 7µm-thick colon cryosections were fixed with 10% buffered formalin for 30 minutes. After fixation, slides were washed, incubated in 3% acetic acid (Carl ROTH) for three minutes, then stained with Alcian Blue (Merck) for 30 minutes, then washed and counterstained with Nuclear Fast Red (Merck) for five minutes. For H&E staining, 7 µm-thick liver cryosections were fixed in methanol for 10 minutes. Slides were left to dry for two minutes, then stained for one minute in hematoxylin, then washed and stained for three minutes in eosin. Both liver and colon slides were dehydrated in ethanol (80%, 95%, 100%), then stained for five minutes in Neo-Clear® (xylene substitute). For immunofluorescent staining, slides were fixed for 10 minutes in acetone, which was left to evaporate, then blocked with 10% FBS (in PBS) for one hour. Slides were stained overnight with the primary antibody (1:200) at 4 °C. Slides were washed with 0.01% Tween-20 (in PBS) the following morning and stained with a secondary antibody (1:250) and DAPI (1:1000) in 1% FBS (in PBS) for an hour at room temperature. Slides were then washed with 0.01% Tween-20 (in PBS). For mounting of slides, 5–6 µL of Tissue-tek coverslipping resin were dripped onto each sample and a coverslip was placed on top.

## ELISA

LPS ELISAs were performed on mouse plasma samples using the ENDONEXT™ EndoELISA® Endotoxin Detection Assay (BioVendor) according to the manufacturers' instructions. Lipocalin-2 was detected in mouse fecal samples using the Mouse Lipocalin-2/NGAL DuoSet ELISA Kit (DY1857-05, R&D Systems) according to the manufacturers' instructions.

## Sodium acetoacetate synthesis and long-chain fatty acid – albumin complexation

Sodium acetoacetate (NaAcAc) was produced in-house by hydrolysis of an equimolar solution of NaOH (Sigma-Aldrich) with ethyl acetoacetate (Sigma-Aldrich). The mixture was heated to 37 °C in a water bath overnight to produce sodium acetoacetate, water, and ethanol. A rotary evaporator with vacuum was used to evaporate out the ethanol and the water and the remaining precipitate was stored at −20 °C until use.

Stearic acid, oleic acid and lauric acid (Sigma) were solubilized at 80 mM in isopropanol (Sigma), by heating to 37 °C for ten minutes. Bovine serum albumin (BSA, Carl Roth) was solubilized in cell culture medium at 2%. Pre-warmed dissolved long-chain fatty acids and pre-warmed BSA were mixed at a 1:1 ratio and complexed in a water bath at 37 °C for an hour. The BSA-long-chain fatty acids and control solutions were filtered through a 0.45 µm sterile filter before use.

NB: When these products were used for experiments, the indicated concentrations were calculated based on the assumption that all

obtained products were pure and all available long-chain fatty acid molecules were complexed to BSA. Thus, the true bio-available concentration of these products remains undetermined, and lower than indicated.

## Cell culture maintenance

HCT116 (CCL-247), HT-29 (HTB-38) and Caco-2 (HTB-37) CRC cell lines were obtained from ATCC and maintained in DMEM-F12 with 10% [v/v] fetal bovine serum (FBS) and 1% [v/v] penicillin/streptomycin. Cell lines were mycoplasma-free and authenticated before their use in this study (STR analysis, DSMZ).

## T cell differentiation assay

Spleens and lymph nodes from young ( < 8-week-old) male and female C57BL/6 J mice were isolated using the CD4$^+$ T Cell Isolation Kit (Miltenyi Biotec) according to the manufacturer's instructions. Single cell suspensions were counted using a Neubauer improved counting chamber (VWR). 200 000 cells per well were dropped onto a 96-well plate pre-coated with anti-CD3 at 5ug/mL per well (incubated for at least three hours at 37 °C) and preloaded with a Th17 cell differentiation cytokine mix (final concentrations: 2 ng/mL TGF-β (R&D systems), 30 ng/mL IL-6 (Miltenyi Biotec), 5 μg/mL anti-IFNγ and 1 μg/mL anti-CD28) and with the treatments of interest at the concentrations indicated in each figure. Three days later, cells were restimulated, stained and analyzed as described above.

## Proliferation assays

For the evaluation of dose-dependent responses to stearic acid, 3000 HCT116, HT-29 and Caco-2 cells were seeded per well in a 96-well plate and were treated with the indicated concentrations of BSA-stearate or a matching control containing cell culture medium, BSA and isopropanol. Confluence was monitored with the IncuCyte imaging system (Sartorius). The IncuCyte ZOOM software (2018B, Sartorius) was used to create a mask for confluence detection.

For the evaluation of the effect of different long-chain fatty acids on tumor cell growth, 6500 HT-29 cells were seeded per well in a 96-well plate. 24 hours after seeding, cells were treated with BSA-stearate, BSA-oleate and BSA-laurate at the indicated concentrations or an equal volume of DMSO as control for 48 hours, in the presence or absence of the caspase inhibitor zVAD (Z-VAD-FMK, InVivoGen). Staurosporine (81590-10, SanBio), a protein kinase C inhibitor, was used as a positive control. Wells were imaged using Cytation10 every 24 h and cell viability was assessed using CellTiterGlo (G9241, Promega) 72 h after treatment according to the user manual.

## RNA isolation from murine tissues and Quantitative Polymerase Chain Reaction (qPCR)

RNA was isolated from mouse tissue samples using the Machery-Nagel Nucleospin RNA Isolation Kit (ThermoFischer Scientific), following the kit manual, after tissue homogenization by bead-beating in the TissueLyzerII (Qiagen) for five minutes, at 40 Hz. The concentrations and purities of the extracted RNA samples were measured using a Nanodrop 2000 spectrophotometer.

cDNA was obtained by reverse transcription using the High-Capacity cDNA Reverse Transcription Kit (Applied Biosystems, Thermofisher Scientific). The Absolute Blue qPCR SYBR Green Low ROX Kit (ThermoFisher Scientific) was used for the amplification. 125 nM of each primer (primer sequences can be found in Supp. Data File 11) and 50 ng of cDNA was used per reaction and run on a 7500 FAST Real time PCR Detection System cycler (Applied Biosciences) set to 95 °C for 15 minutes, followed by 40 cycles of 95 °C for 30 seconds, 60 °C for 30 seconds and 72 °C for 30 seconds. Expression levels of the genes of interest were normalized against the reference genes β-actin and GAPDH using the qbase+ software, in accordance with the MIQE guidelines[48].

## RNA extraction from cell pellets and sequencing

RNA was extracted from treated cell pellets using the miRNeasy Mini Kit (#74106, Qiagen). DNA was digested using RNase-free DNase (RNase-free DNase Set, #79256, Qiagen). For library preparation, 1 μg of RNA was poly(A) selected, fragmented and reverse transcribed by using the Elute, Prime, Fragment Mix (Illumina). 1 μg of RNA was poly(A) selected and fragmented by using the Elute, Prime, Fragment Mix (Illumina). Reverse transcription, End repair and adapter ligation and library enrichment were performed as described in the Low Throughput protocol of the TruSeq RNA Sample Prep Guide. RNA library quantity and quality were assessed with the Fragment Analyzer (Agilent). RNA libraries were sequenced as 50 bp paired-end on Illumina NextSeq2000 sequencer at the LCSB genomics platform (RRID: SCR_021931).

## RNA data processing and analysis

FastQ files were processed using a snakemake workflow which can be found here: https://gitlab.lcsb.uni.lu/aurelien.ginolhac/snakemake-rna-seq. Tag version: 0.5.4. Container software set: https://hub.docker.com/r/ginolhac/snake-rna-seq tag 0.8. Briefly, reads were trimmed with AdaptorRemoval with a minimum length of the remaining reads set to 35 bp (version 2.3.3[49,50]). Trimmed reads were mapped to human reference genome GRCh38 (from Ensembl release 102) using STAR v2.7.11a[51]. Quality control was performed using fastqc (version 0.12.1) and fastq Screen (version 0.15.3[52]) and summarized using multiqc (version 1.17). Read counts were extracted using Rsubread::featureCounts v2.16.0[53] and reads with counts below 10 were removed prior to differential expression analysis using DESeq2 version 1.42.0 with apeglm to preserve large log2-fold changes v1.24.0[54]. The identified batch effect (biological replicate) was removed using ComBat-Seq implemented in the R package sva[55] and differential gene expression analysis performed using DESeq2[56]. Log2 fold change values were shrunken using ashr[57]. Genes were pre-ranked by the log2 fold-change values and geneset enrichment analyzes performed using the GSEA() function implemented in the clusterProfiler R package. Hallmark genesets were retrieved using the msigdbr R package and the Chen et al. genesets[58]. Heatmaps were rendered using the Complex-Heatmap R package. TCGA COAD and READ data (data release v36) were retrieved from the GDC portal and the expression data from patients providing paired distant normal and primary tumor samples filtered out ($n = 50$ patients). Geneset expression scores for TCGA and our in-house RNA-Seq dataset were calculated using the R package singscore[59,60] and TPM expression values.

## DNA isolation from fecal samples (mouse, human)

Fecal DNA was isolated using the DNeasy PowerSoil DNA Isolation Kit (Qiagen), according to the kit manual, but with the following modification - the sample was homogenized using the TissueLyzerII for five minutes at 40 Hz instead of by vortexing. DNA extracted from human fecal samples was additionally subjected to processing through the DNA Clean and Concentrator Kit (Zymo Research). The concentrations and purities of the extracted DNA samples were measured using a Nanodrop 2000 spectrophotometer.

## 16S rRNA amplicon sequencing and WGS

For 16S rRNA sequencing, 5 ng of DNA extracted from human or mouse fecal samples was subjected to amplicon sequencing of the V4 region of 16S rRNA genes using primers 515 F and 805 R (primer sequences can be found in Supp. Data File 11). Samples with low microbial biomass were excluded. Sequencing was performed on a MiSeq instrument (Illumina) using $2 \times 300$ base pair read lengths.

For WGS, 200 ng of DNA extracted from human fecal samples was used for metagenomic library preparation using the Next Generation Sequencing DNA Library Prep Kit (Westburg), according to the user manual. 100 ng of DNA extracted from mouse fecal samples

was used for metagenomic library preparation using the FX DNA Library Kit (Qiagen), according to the user manual. The average insert size of libraries was 450 base pairs for human fecal samples and 530 base pairs for mouse fecal samples. Sequencing was performed on a NextSeq500 (Illumina, for human fecal samples) or on a NextSeq2000 (Illumina, for mouse fecal samples) instrument using $2 \times 150$ base pair read lengths.

All prepared libraries were quantified using the Qubit® dsDNA BR Assay Kit (Thermofisher Scientific) and Qubit® 3 Fluorometer (ThermoFisher Scientific) and sample quality was assessed using the Bioanalyzer High Sensitivity DNA Analysis Kit (Agilent Technologies).

### 16S data analysis
Raw 16S rRNA gene sequences (FASTAQ files) were cleaned, processed and merged up to an amplicon sequence variant (ASV) count table using the DADA2 pipeline[61]. SILVA train sets (versions 132 (engraftment of 5HD mix in mice) to 138 (community changes over time)) were used for taxonomy inference. Samples with no reads, ASVs with no identified phylum or class and taxa present in less than 5% of samples were excluded where necessary. Data was aggregated to genus level and relative abundances were computed, or centered log-ratio (CLR) transformation[62] was performed. Differential analysis and graphical representation were performed in R, using the packages from Bioconductor (DESeq2[56,63], phyloseq[64]) and tidyverse (ggplot).

### WGS data analysis for human fecal samples
Raw genome sequences (FASTQ files) were cleaned using the Sunbeam pipeline[65]. Function profiling was performed with HUMAnN Version 3.0[66] using the quality-controlled and host-decontaminated FASTQ files obtained from Sunbeam. Graphical representation was performed in R, using the packages from tidyverse, broom and RColorBrewer.

### WGS data analysis for mouse fecal samples
The Integrated Meta-omic Pipeline (IMP; v3 - commitID #6f1badf7)[67] was used to process paired forward and reverse reads using the built-in metagenomic workflow. The workflow includes pre-processing, assembly, genome reconstruction and functional annotation of genes based on custom databases in a reproducible manner. De novo assembly was performed using the MEGAHIT (version 2.0) assembler. Default IMP parameters were retained for all samples. MetaBAT2 and MaxBin2 in addition to binny[68] were used for binning and genome reconstructions, i.e., MAGs. Subsequently, a non-redundant set of MAGs was obtained using DASTool v1.1.461 with a score threshold of 0.7 for downstream analyzes, and those with a minimum completion of 90% and less than 5% contamination as assessed by CheckM v1.1.3. Taxonomy was assigned to the MAGs using the extensive database packaged with gtdbtk v2.0.1. In addition, Kraken2 implemented in IMP was used for taxonomic assignment of reads. For the analyzes of functional potential from the assembled contigs, open-reading frames were predicted from contigs using a modified version of Prokka that includes Prodigal gene predictions for complete and incomplete open-reading frames. The identified genes were annotated with a hidden Markov models (HMM) approach, trained using an in-house database including KOs derived from the Kyoto Encyclopedia of Genes and Genomes (KEGG) database[69–71].

Differential analysis was performed using the package DESeq2[56] and using Wilcoxon rank sum test performed on CLR-transformed Kraken2 reads counts. Non-metric dimensional scaling (NMDS) analyzes were performed on robust Aitchison distances calculated from Kraken2 read counts using the R package *vegan* (version 2.6-6.1). For further analysis as well as graphical representation R packages *ggplot2, ggpubr* (version 0.6.0), *tidyverse, gghx4* and *RColorBrewer* (version 1.1-3) were used.

### Human fecal sample and mouse fecal pellet homogenization for metabolite extraction
Circa 1 g of frozen human fecal material was milled in a cryogenic grinder (6875D Freezer/Mill®, SPEX® SamplePrep, Instrument Solutions) reducing the samples to a fine powder. Homogenized fecal powders were resuspended in 500 μL of MilliQ® water, then centrifuged at maximum speed.

MilliQ® water was added to mouse fecal pellets at a 1:16 dry weight to water ratio, in tubes containing ceramic homogenization beads. Samples were homogenized at 6000 rpm, for two 30-second-long cycles at 4 °C, in a Precellys®24 Homogenizer (Bertin Corp.), incubated at 4 °C for ten minutes, then centrifuged at maximum speed.

Supernatants were collected for further downstream processing.

### Metabolite extraction
An internal standard mix (ISM) was added to the samples (human fecal samples, murine fecal samples and murine plasma samples). Proteins were precipitated using methanol (4 parts per 1 part sample extract) and pelleted before supernatants were collected. Phase separation was achieved by addition of chloroform, aliquots of the polar and non-polar phase were transferred into GC-MS vials and evaporated in a rotary vacuum concentrator at 4 °C overnight for subsequent GC-MS analysis. Phospholipids were removed from sample extracts before drying for subsequent HILIC LC-MS analysis.

After reconstitution of the samples, hydrophilic interaction chromatography (HILIC) was performed on a Thermo Vanquish UHPLC coupled to a Thermo Q Exactive HF mass spectrometer (Thermo Fisher Scientific) equipped with SeQuant ZIC-pHILIC $150 \times 2.1$ mm columns (Merck Millipore) using 20 mM ammonium acetate in water as mobile phase A and 20 mM ammonium acetate in 90% acetonitrile as mobile phase B.

Before untargeted GC-MS analysis of murine plasma samples, polar metabolites were derivatizedusing a multi-purpose sampler (Gerstel). Samples were measured on a 7890B GC (Agilent Technologies) coupled to a 5977A mass selective detector (Agilent Technologies).

Detailed descriptions of the analytical methods are provided in the Supplementary Information File.

### Metabolite extraction from murine fecal samples for short-chain fatty acid (SCFA) quantification
For targeted and absolute quantitative SCFA detection in mouse fecal samples, a dilution series of a volatile free acid mix (VFAM, Supelco®, Sigma-Aldrich) and separate quality control samples were prepared and analyzed alongside the test samples. SCFA extraction was performed by adding internal standard 2-ethylbutyric acid (Sigma-Aldrich) in methanol, MilliQ® water and hydrochloric acid (Sigma-Aldrich) to all samples. After 15 minutes of agitation at 15 °C in the Thermomixer (Eppendorf), two subsequent diethyl ether extractions were performed. The decanted supernatants were pooled, then aliquoted in triplicates. SCFA derivatization was performed by adding N-Methyl-N-(tert-butyldimethylsilyltrifluoro acetamide) w/1% tert butyl-dimethylchlorosilane (MTBSTFA, Interscience). The samples were analyzed on an 8890 GC (Agilent Technologies) coupled to an 5977B MSD (Agilent Technologies).

Detailed descriptions of the analytical methods are provided in the Supplementary Information File.

### Murine plasma bile acid (BA) quantification by ultra-high performance liquid chromatography – tandem mass spectrometry (UHPLC-MS/MS)
10 μL of plasma were used for bile acid (BA) quantification. BAs were analyzed using UHPLC-MS/MS, consisting of an ExionLC (Sciex) coupled to a QTrap 5500 mass spectrometer (Sciex). Electrospray ionization was performed in the negative ionization mode. Chromatographic

separation was run on an ethylene bridged hybrid (BEH) C18 column (2.1 × 100 mm, 1.7 μm, Waters). The mobile phases consisted of water, containing 0.1% formic acid including 5 mM ammonium acetate and acetonitrile (ACN, Carl ROTH). The analytes were separated by a gradient elution. BA detection was performed in the multiple reaction monitoring (MRM) mode. Standards for all shown BAs, as well as the deuterated BA internal standard (IS) substances d4-CA, d4-GCA and d4-TCA were purchased from Sigma-Aldrich Chemie GmbH (Taufkirchen, Germany), Avanti Polar Lipid (Alabaster, USA) and Steraloids (Newport, USA). Further details can be found in García-Cañaveras et al.[72] and in the Supplementary Information File.

### Metabolomic profiling data pre-processing of untargeted LC-MS datasets

Raw data files obtained through LC-MS of human stool and murine plasma and stool samples were processed in TraceFinder (version 5.1.203.0) for peak identification and annotation. Three different in-house libraries, generated with reference standards, as well as one commercially available library, mzCloud Offline for mzVault 2.3_Omics_2020A.db, in the Advanced Mass Spectral Database (AMSD) (HighChem, from ThermoFischer Scientific), were used for peak annotation, with the adduct formulas $[M + H]^+$ and $[M-H]^-$. Annotated features and integration tables were exported for post-processing.

### Metabolomic profiling data pre-processing of untargeted GC-MS datasets

Raw data files obtained through GC-MS murine plasma and stool samples were processed in MetaboliteDetector[73] (version 3. 220190704) for peak identification and annotation. An in-house library, generated with reference standards was used for peak annotation and analyte identification. Deconvolution settings were applied as follows: peak threshold = 5, minimum peak height = 5, bins per scan = 10, deconvolution width = 5 scans, no baseline adjustment, minimum 15 peaks per spectrum and no minimum required base peak intensity. The data was normalized using the response ratio of the integrated peak area of each metabolite and the integrated peak area of the IS, as described in ref. 74. Annotated features and integration tables were exported for post-processing.

### Metabolomic profiling post-processing data analysis

For murine plasma metabolites identified in the SPF experiment, data was manually curated based on the pooled sample, spiked with the ISM. Features, which could not be confirmed through MS2 data (from the commercially available database[75]) or through the ISM were filtered out. Metabolites with missing (N/A) values were also filtered out. Single metabolite intensities were normalized to the total metabolite area and corrected using the mean value of total metabolite values from all samples. LC-MS and GC-MS data were unified, and duplicate metabolites were filtered out, based on reliability and consistency of the peak areas.

Human stool sample data was similarly processed (but without the element of data unification between methods).

For murine plasma metabolites identified in the GF experiment (GC-MS), metabolites with missing (N/A) values were filtered out.

Murine stool SCFA content was quantified based on an IS and using the MassHunter Quantitative Analysis Software (version 10.2.733.8). In the LC-MS and GC-MS datasets, metabolites with over 20% missing (N/A) values were filtered out. Data from murine stool untargeted LC-MS, GC-MS and SCFA quantification was unified and duplicate metabolites were filtered out, based on reliability and consistency of the peak areas.

Univariate statistical analysis with Benjamini-Hochberg false discovery rate (BH FDR, performed for all datasets), principal component analysis (PCA), scaling, centering and graphical representation in the form of a heatmap (ComplexHeatmap R package[76]) or log2 fold changes were performed in R version 4.2.1 and Rstudio version 1.4.1717. For

data scaling and centering, the standard deviation was set to a value of 1 and the mean value was set to 0.

### Total and free fatty acid extraction

Powders of mouse diet samples were prepared by placing samples in 7 mL Precellys® tubes (VWR) containing 10 steel beads and homogenization at 9000 rpm for 3 cycles of 30 seconds at 4 °C to. For extraction of total and free fatty acids, mouse diet powders and mouse fecal samples were homogenized in Precellys® tubes (VWR) containing 600 mg ceramic beads and MeOH + ISM (final concentration 2 μg/mL) in a sample to weight ration of 1:30. Non-polar fatty acids were extracted using methyl-tert-butyl ether (MTBE) and $H_2O$ to induce phase separation. For quantification of free fatty acids, aliquots of the non-polar phase were transferred into glass vials, solvents were evaporated, and samples were kept until data acquisition. For quantification of total fatty acids, dried non-polar extracts were dissolved in isopropanol and 0.4 M NaOH was added to perform alkaline hydrolysis. After incubation for 30 min at 37 °C, samples were acidified with 2 M HCl and fatty acids were extracted using hexane.

Plasma fatty acids were extracted from 8 μL mouse plasma mixed with ACN/$H_2O$ (4 + 1) + ISM (final concentration 2 μg/mL) using the same workflow with the exception that an acidic hydrolysis was performed prior phase separation using 5 M HCl.

Dried total fatty acid extracts were reconstituted, filtered using Phenex-PTFE syringe filters and analyzed on an Agilent 1290 LC coupled to an Agilent 6560 Q-TOF MS system equipped with a Dual Agilent Jet Stream ESI source using Agilent Mass Hunter LC/MS Data Acquisition (version B.09.00, Build 9.0.9044.0) for data acquisition.

Detailed descriptions of the analytical methods are provided in the Supplementary Information File.

### Pre-processing of total and free fatty acid datasets

Raw data obtained through fatty acid LC-MS profiling were processed in Agilent Mass Hunter Profinder (ver 10.0 SP1, Build 10.0.10142.1). Target compounds were identified by exact mass (mass error ± 5 ppm), isotopic pattern and retention time (± 0.15 min) matching (Batch Targeted Feature Extraction). Semi-quantification was based on integrated peak area of the deprotonated target compound and normalization to internal standards.

### Statistics and reproducibility

Detailed descriptions of statistical analysis are provided in the above "Methods" sections. All other data analysis, statistical analysis, and graphical representation, which is not described in the above "Methods" sections, was performed using Graphpad Prism version 9.4.1, and is included in the figure legends. For in vitro studies a minimum of technical triplicates per condition were used and each experiment was verified in at least three independent experiments to ensure robust and reliable outcomes. Specific number of repetitions for each experiment is provided in the figure legends of the manuscript. For in vivo work, a power calculation was used to determine group sizes in liaison with a bio-statistician. Mice were randomly allocated to experimental groups prior to the start of the experiments. Specific number of animals used per experiment and number of experiments performed is provided in the figure legends of the manuscript. No data was excluded unless clear technical issues were identified. Order of acquisition of all Mass Spec samples and DNA sequencing samples was randomized during the analysis to avoid bias from ex. instrument drift. Analyzes of in vivo samples (ex. tumor counts) were blinded and analyzed by two researchers independently. IDs were uncovered after final result was obtained.

### Reporting summary

Further information on research design is available in the Nature Portfolio Reporting Summary linked to this article.

## Data availability

Raw sequencing data files for 16S gene sequencing from murine fecal samples are available at the ENA's sequence archive under the accession PRJEB70917 (GF) and PRJEB70920 (SPF). Raw sequencing data files for 16S gene sequencing from human stool samples are available at the ENA's sequence archive under the accession PRJEB70932. Raw sequencing data samples for mouse stool samples are available at the ENA's sequence archive under the accession PRJEB66281. Raw sequencing data files for RNAseq from LCFA-treated HT29 cells can be accessed under ArrayExpress accession number E-MTAB-14429. Raw sequencing data files for WGS from human stool samples have been filtered for human reads and filtered data are available at the ENA's sequence archive under the accession number PRJEB84188. Metabolomics data processed with Tracefinder are available at the NIH Common Fund's National Metabolomics Data Repository (NMDR) website, the Metabolomics Workbench, https://www.metabolomicsworkbench.org[77]. Untargeted LC-MS profiling data of donor stool samples have been assigned Study ID ST003681, untargeted LC-MS profiling data of fecal samples from SPF diet experiment have been assigned Study ID ST003679, untargeted LC-MS profiling data of plasma samples from SPF diet experiment have been assigned Study ID ST003683 and untargeted LC-MS profiling of fecal samples from CMT experiment have been assigned Study ID ST003680. The data can be accessed directly via its Project https://doi.org/10.21228/M8HZ6G. Source data are provided with this paper.

## Code availability

Details on workflow and codes/scripts can be found here: https://gitlab.com/uniluxembourg/fstm/dlsm/mdm/tsenkova_et_al_2024.

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

## Acknowledgements

We acknowledge the contributions of the following people in the context of this project and thank them for their work: Dominik Ternes, Rubens Begaj, Martin Nurmik, Mohaned Benzarti and Monica Gabola (MDM group, DLSM) for assistance during organ processing at experimental endpoint; Ysaline Bossicard (MDM group, DLSM) for tissue cryosectioning. We thank the LCSB Metabolomics and Lipidomics Platform (RRID:SCR_024769) for their technical and analytical support with our metabolomics samples, especially Xiangyi Dong, Floriane Gavotto and Christian Jäger. We further thank Aurélien Ginolhac (DLSM) for contributions to data pre-processing; Pau Perez Escriva (DLSM) for bacterial culturing; Djalil Coowar, Jennifer Behm, Julie Virolle, Cloe Ordrenneau and all other members of the Animal Facility Team (LCSB); Sara Gomez and Eliane Klein (DLSM) for maintenance of organoid and cell cultures; Rashi Halder for fecal sample 16S and WGS sequencing. The sequencing experiments presented in this paper were carried out at

the LCSB sequencing platform (RRID: SCR_021931) at the University of Luxembourg. We also thank the donors who contributed their samples to the MuSt cohort, as well as Christelle Bahlawane (CIEC), Eve Herkenne (CIEC), Laetitia Garcia (CIEC), Ingrid Vreja (CIEC), and nurses from CIEC for contributing to this cohort. We thank Dr. Barbara Kofler for her feedback and discussions during the course of the project. The experiments presented in this paper were carried out using the high-performance computing facilities at the University of Luxembourg (https://hpc.uni.lu)[78]. BioRender (BioRender.com) was used for graphical illustration. This work was supported by the Luxembourg National Research Fund (FNR) (grant nos. CORE/C16/BM/11282028 (E.L.), PoC/18/12554295 (E.L.), AFR 17103240 (C.G.), PRIDE17/11823097 (M.T., M.K., L.d.N.) and CORE/15/BM/10404093 (P.W.)), by the Luxembourg National Research Fund and the Fondation Cancer Luxembourg (grant no. CORE/C20/BM/14591557 (E.L.)), as well as by the Fondation du Pélican de Mie and Pierre Hippert-Faber under the aegis of the Fondation de Luxembourg ('Pelican Grant'; M.T. and M.M.), a FNRS-Télévie grant to M.M., no. 7.4565.21-40007364), an Internal Research Project at the University of Luxembourg (MiDiCa—integrated analysis of the effects of microbiome-diet interactions on human colorectal adenocarcinoma enterocytes; E.L., P.W. and S.H.), the Fondation Cancer and the Fondation Kriibskrank Kanner Luxembourg (V.I.P), the Action LIONS Vaincre le Cancer Luxembourg and a European Research Council grant under the European Union's Horizon 2020 research and innovation program (grant agreement no. 863664 to P.W.). This project was also supported by the Doctoral School in Science and Engineering (M.T., M.K., M.M. and L.d.N.) and the Department of Life Sciences and Medicine at the University of Luxembourg. The funders had no role in study design, data collection and analysis, decision to publish or preparation of the manuscript. This work is supported by NIH grant U2C-DK119886 and OT2-OD030544 grants as metabolomics data has been uploaded at the NIH Common Fund's National Metabolomics Data Repository (NMDR) website.

## Author contributions

Conceptualization was done by M.T., V.I.P., and E.L. Methodology was devised by M.T., V.I.P., and E.L. Clinical methodology was developed by C.d.B. Formal analysis was carried out by M.T., V.I.P., M.B., M.K., S.B.B., L.d.N., D.C., D.H., A.G., E.K., and M. Schmitz. The investigation was carried out by M.T., V.I.P., M.K., M. Schmoetten, M. Schmitz, F.R., M.M., C.G., D.C., and L.S. Data curation was done by M.B., M.K., E.K., A. G., S.B.B., J.M., and L.D.N. Visualization was done by M.T., M.B., V.I.P., M.K., D.C., A.G., and E.K. The original draft was written by M.T. Review and editing of the draft was done by M.T., M.B, E.K., L.S., P.W., and E.L. Project administration was carried out by E.L. Funding was acquired by E.M., J.M., P.W. and E.L.. Finally, S.H., T.C., C.L., P.W., and E.L. supervised the project.

## Competing interests

The authors declare no competing interests.
