## [Transparent Peer Review file · Nature Communications]

Ketogenic diet suppresses colorectal cancer through the gut microbiome long chain fatty acid stearate

Corresponding Author: Professor Elisabeth Letellier

Version 0:

Reviewer comments:

Reviewer #1

(Remarks to the Author)

The authors conduct several sets of mouse experiments and identify plausible mechanisms of KD in anti-tumor effects. The overall findings are strong and well-confirmed through relevant in-vitro/in-vivo experiments. However, some aspects of the study design appear inappropriate or unnecessary, and some conclusions are not persuasive, as mentioned below:

Study design: The author should provide a clear statement and explanation for why both GF and SPF mice were utilized.

Line 97-99: Since inflammation played a major role in determining the anti-tumorigenic effect, the author should have concluded with a solid observation regarding whether the reduced inflammation resulted from the Ketogenic diet intervention or was a direct consequence of reduced tumor development. Conducting correlational analysis between inflammation markers and tumor number (or burden) could assist in addressing this question.

Line 103-105: The authors mention selecting a specific CRC donor rather than mixing stool samples from multiple donors due to high heterogeneity in microbial profiles. However, it is suggested that obtaining several samples from different donors and creating a mixture could reduce heterogeneity and provide a more representative microbiome for FMT. The authors didn't observe significant differences from mice that received FMT from a CRC donor and concluded that the microbiome of CRC patients did not exhibit a higher potential for CRC induction. However, if they had selected another CRC donor, it is likely to have yielded very different results, and the authors would have reached different conclusions. For this reason, the results from mice that received FMT from a single CRC donor seem inappropriate and nonreproducible for measuring the effects of the microbiome on CRC induction and should be removed from the manuscript.

Line 105-113: Overall, the results and conclusions derived from these results are contradictory and not persuasive. For instance, in the low-grade inflammatory model, no significant differences in colonic tumor burden and immune cell profiles are observed. However, the conclusion drawn is that "KD exerts an anti-tumorigenic effect in CRC, potentially by modulating the immune system." This raises questions about why the Ketogenic diet did not affect tumor burden and immune cell profiles in the inflammatory model and how such a conclusion was reached with non-significant results.

Line 131: The authors tested the effect of palmitic/oleic acid treatment on Th17 differentiation. Whereas, stearic acid is considered the most important fatty acid in reducing tumorigenesis. It is questioned why the authors did not test this fatty acid in Th17 differentiation.

Line 141-144: It is unintelligible why the authors set this study design (utilizing open-top cages in a gnotobiotic flexible-film isolator to allow for microbial exchange between groups) to confirm the mediating effect of the microbiome on the anti-cancer effects of KD. For this type of research question, fully separate housing to inhibit microbial exchange between groups would be better to observe the modulation effect of the gut microbiome. As such, it remains questionable whether mice having different microbiome profiles will lead to different results in anti-cancer effects.

Additionally, according to the study design figures, it seems the authors performed FMT several times throughout several weeks. However, FMT is ideally or mostly conducted at the very early stages of experiments for microbiome transplantation to observe the long-lasting effect of FMT.

Line 181: Various *Alistipes* spp. were differentially abundant, but it is unclear whether this bacterium is associated with the

anti-cancer effect throughout the manuscript. The author should evaluate and discuss the implications of the enrichment of this bacteria and whether it contributes to the anti-cancer effect.

Line 189-207: Given the considerable differences in the microbial profiles of CMT-donor mice and recipient mice, significant differences in their functional profiles are already expected. Therefore, the description of the functional profiles should focus on finding consensus functional profiles between CMT-donor and recipients, despite significant differences in profile, rather than describing it as if it 'maintained'.

Line 333-335: The authors mention the difference between responder and non-responder, which is based on their different microbial profiles. However, since all the mice in this study received FMT, and the basic assumption of FMT is that the recipient will share a similar microbial profile with the donor and other mice receiving the same FMT, the purpose and assessment of microbiome effects on health are unclear. If they indeed had different microbial profiles after FMT, then what is the purpose of FMT, and how can the microbiome effects on health be assessed?

Methods: 1) The methods for extracting genomic DNA from fecal samples for sequencing and preparing stool/plasma samples for metabolomic profiling are omitted. 2) The specific 16S region used for experimentation in the '16S rRNA sequencing data analysis' section is not mentioned. 3) Also, DADA2 generates ASVs not OTU. 4) The correct name of the taxonomy database name is 'SILVA'. 5) There are profound differences between SILVA versions 132 and 138. Why did the authors utilize both versions instead of just the current/latest one?

Figure 5d: All the ketogenic mice have either no tumors or very few (seemingly 1-2), making it doubtful whether the Ketogenic diet truly has such a strong effect in inhibiting tumorigenesis.

Ext. Figure 4e-f: The authors provided differences in BAs but did not describe the significance of these differences. It is better to provide implications rather than just showing differences.

Ext. Figure 6: The group labels 'Standard GF' and 'Ketogenic GF' do not match with the group labels used in the Results section (SD-CMT and KD-CMT). These labels may lead to misinterpretation as they indicate germ-free status. It is recommended to change the labels to match those used in the Results.

Reviewer #2

(Remarks to the Author)

Comments and suggestions for authors

In this manuscript, Tsenkova and colleagues find that stearate, a long-chain fatty acid that is produced after ketogenic diet feeding, can reduce colorectal cancer burden. The authors describe, in great detail, the phenotypic, metabolic, and gut bacterial/immune composition changes that occur in mouse models under ketogenic diet. This is an extensive study describing how the ketogenic diet might influence responses to CRC, and the findings presented in this paper can fill the gap between KD-altered gut microbiota and CRC development.

However, additional evidence is needed to support the conclusions drawn by the authors.

Major comments:

1. The manuscript starts by detailing many correlations with KD and gut/immune phenotype, but do not connect these final observations of Stearate upregulation with initial observations, keeping in mind that a major conclusion of the paper is that the KD-derived stearate reduces CRC tumor burden, this should be connected.

2. Why would reduced populations of immune infiltrates (ED Fig 2) cause or be a result of reduced tumor burden? Is this reduced tumor burden merely reflective of the increased Stearate in the circulation from KD that is found to suppress CRC?

3. Does oral administration of stearate reduce CRC burden in the AOM/DSS model?

4. Th17 downregulation is noted in the mouse models. What is the role of Th17 CD4+ T cells in the control of CRC tumors? If their reduction/downregulation is important for KD-mediated tumor control, is there a benefit in tumor control from using CD4-T cell depletion in the AOM/DSS CRC model in SD-mice?

5. On this same note, it was mentioned (lines 125-127) that KD-derived ketone bodies - acetone and Na-acetoacetate - reduced Th17 differentiation. How does stearate regulate Th17 differentiation and in general, immune composition in these mice?

1. Following this point, does KD-CMT induce Th17 reduction, or general immune cell reduction, as shown in the KD-mice of ED Fig 2d/e? Clearly, this is an important connection to link the role of the microbiome with any speculation of immune-mediated tumor control.

2. The authors tried to assess KD suppress CRC via modulating inflammation by using AOM model instead of AOM/DSS. However, in addition to being an inflammatory agent, DSS treatment also compromises epithelial barrier integrity, changes microbial composition, which allowing the bacteria and bacteria products invade into underlying tissue. In combination of suggestion 6 and 11, the results are not robust enough to state the anti-tumorigenic effect of KD in CRC is mediated by modulating the immune system (line 112).

Minor comments:

3. No figure legends.

4. In Line 96, the authors state calprotectin level was reduced in the spleen of GF KD-fed mice. Calprotectin is a common diagnostic evaluation of IBD patients, but usually, the fecal calprotectin is used. How's the fecal calprotectin level in this study? what is the rationale for using spleen calprotectin level to indicate gut inflammation?
5. The schema of Figure 1a is not clear, for example, there appears to be overlap between antibiotic treatment and 5HD gavage.
6. Line 211, in this part, the authors identified that the microbial metabolite stearate is elevated by KD and exhibits anticancer effects. This is a very interesting finding, and it is also the major conclusion of this paper. It is too long to be one section. It would be better to separate this section into three parts: 1. Fecal metabolic profile analysis indicates an increase in stearic acid level in KD-fed mice. 2. Elevated stearic acid level is associated with changes in bacterial taxa that related to fatty acids producing and/or consuming. 3. In vitro treatment or in vivo supplementary of stearic acid can suppress CRC progression.
7. Line 102 What is the rationale for using CRC patient FMT in AOM model if it's not mentioned in the AOM/DSS model? Would there be any difference between CRC-FMT and 5HD-FMT in AOM/DSS model?
8. Is the low-inflammatory model in ED Fig 3 capable of generating sufficient colonic tumors for analysis? i.e., is it possible that we do not see statistical differences in SD vs KD simply because the model is not capable of generating sufficient colonic tumors?
9. Please also show the antibody depletion efficiency if you collected the stool after antibiotic treatment and before 5HD gavage. Additionally, what is the implantation efficiency? It would be beneficial to compare the composition of bacteria in the input 5HD oral gavage samples and stool samples after CMT.
10. It would be better to have a SPF-stool-FMT control or germ-free stool-FMT control for GF group and SPF group, respectively.
11. Line 93 and extended Fig. d and e are not clearly labeled and described. Were the samples from GF or SPF mice? what methods were used to quantify the immune cell profiles? CD4+ and CD8+ T cell populations, which are two major anti-tumor immune cells, were decreased in KD-fed mice. Could you explain the possibility?
12. Line 103, the authors mentioned a single CRC donor was used for colonization because of dissimilarities of microbial profiles. What is the principle to choose this one CRC donor rather than others to perform FMT in this study?
13. Does the // in Extended Data Figure 3a mean a discontinued diet treatment? If so, please explain.
14. In Line 107, according to extended Data Fig. 3f, there are differences in immune cell profiles, eg. MLN macrophage population is different among GF, 5HD and CRC.
15. No statistical analysis of Extended Fig. 4h. 10 mM, 50 mM and 100 mM of BHB are high doses for in vitro treatment. Could the lack of effect of BHB be due to the high concentration? What was the viability of the cells?
16. Line 116, this whole section – KD induces a distinct host metabolic profile, is not associated with other parts in this paper. It is known that KD can alter metabolic profile.
17. Mucus layer thickness (ED Fig 5a) is clearly trending higher in KD than SD. It is unclear why this is not significant (i.e., if different n(mice) are causing non-statistical significance).

Version 1:

Reviewer comments:

Reviewer #1

(Remarks to the Author)

(Remarks on code availability)

Reviewer #2

(Remarks to the Author)

Thank you for revising the manuscript and for addressing the comments from the first round of review. We appreciate the effort you have put into revising the manuscript. Overall, the manuscript has improved significantly and the authors addressed most of the concerns raised by the reviewers.

Major comments

1. & 2. The authors added experiments to investigate the effects of a stearic acid-rich diet on humanized gut microbiome-transplanted germ-free mice in an AOM/DSS model, finding a reduced tumor burden accompanied by a decrease in Th17 cells. KD diet-treated mice show slightly better suppression of tumor burden (Fig. 1c vs. New Fig. 6e.) Can the absolute concentration of stearate be measured and compared between KD diet-treated mice and stearic acid-rich diet-treated mice? Additionally, please add "GF" in New Figure 6a to indicate that germ-free mice were used in this figure.

4. In the authors' response "In our study, KD-fed mice displayed decreased Th17 cells and associated cytokines, suggesting that the ketogenic diet reduces this immune cell subtype...". However, the results showing reduced cytokines are not presented in the paper. Please clarify this point.

6. "We observed a reduction in IL17 levels, consistent with our previous in vivo findings in KD-fed mice." These data of IL17 levels in KD-fed mice could not be found.

Minor comments

Typographical issue: Please correct "C57BL6/J" and "C57BL6/NTac" (eg. Fig.1 a and Fig. 2a) to C57BL/6J and C57BL/6NTac.

(Remarks on code availability)

We would like to thank all the reviewers for their valuable comments. Below, you will find a point-by-point response to their comments. We have additionally submitted a revised version of our manuscript, with all changes highlighted. New sentences corresponding to new results are marked in yellow, while updates to existing text for clarity or repositioning are highlighted in blue in the revised manuscript. Additionally, we have updated the author list to acknowledge the contributions of different researchers during the revisions. Lastly, due to the new results presented in the manuscript, we would like to request a title change that better reflects the current findings of the study. We propose the following title “The gut microbiome-linked long chain fatty acid stearate suppresses colorectal cancer”, as we have now added new results, in particular, a new Figure (Figure 6) in which we assessed the in vivo role of the LCFA stearate, as requested by the reviewers.

Reviewer #1

The authors conduct several sets of mouse experiments and identify plausible mechanisms of KD in anti-tumor effects. The overall findings are strong and well-confirmed through relevant in-vitro/in-vivo experiments. However, some aspects of the study design appear inappropriate or unnecessary, and some conclusions are not persuasive, as mentioned below:

Study design: The author should provide a clear statement and explanation for why both GF and SPF mice were utilized.

We thank the reviewer for this comment. We utilized both germ-free (GF) and specific pathogen-free (SPF) mice for our study. SPF mice possess a fully functional immune system, which is critical for mediating responses in the context of CRC. In contrast, GF mice do not possess a fully mature immune system, limiting some conclusions on the immune cell composition. However, combined with FMT, they offer the advantage of engrafting a human microbiome without competition from an existing microbiome, as would be the case with SPF mice. This approach eliminates the need for antibiotics, which could introduce trace resistance mechanisms and potentially affect the successful engraftment of the human microbiome, ensuring a more controlled and accurate assessment of the human microbiome's impact on CRC. Thus, both models are essential to address our research questions because they provide complementary insights.

Updated text in lines 88-91:

We utilized both germ-free (GF) and specific pathogen-free (SPF) mice to leverage the fully functional immune system of SPF mice for mediating CRC responses and the unique advantage of GF mice allowing engraftment of a human microbiome into mice in the absence of competition with a resident mouse microbiome.

Line 97-99: Since inflammation played a major role in determining the anti-tumorigenic effect, the author should have concluded with a solid observation regarding whether the reduced inflammation resulted from the ketogenic diet intervention or was a direct consequence of reduced tumor development. Conducting correlational analysis between inflammation markers and tumor number (or burden) could assist in addressing this question.

The AOM/DSS model is a well-established inflammation-driven model characterized by a pro-inflammatory environment, where immune cells like macrophages, neutrophils, and Th17 cells are pivotal in promoting both inflammation and tumor development. Based on the reviewer's suggestion, we now detected S100A8 expression in the colon rather than the spleen. Our findings revealed no significant differences between the standard diet (SD) and ketogenic diet (KD) groups (both GF and SPF), indicating similar overall inflammation markers (New Extended Figure 2 g and h). While determining whether reduced inflammation precedes reduced tumor burden would require multiple time-point analyses, such experiments are challenging to execute. Instead, we focused on specific immune cell subtypes known to drive tumor progression in the AOM/DSS model. Th17 cells, in particular, are clinically relevant in CRC and play a critical role in this model. Elevated levels of IL-17A and ROR γ t in CRC patients' tumor tissues are associated with a significant reduction in disease-free survival, highlighting the pro-tumorigenic role of Th17 cytokines^{1,2}. This is further supported by evidence from IL-17A-deficient mice in the AOM/DSS model, which show reduced expression of IL-6, STAT3, IFN γ , and TNF α , leading to milder colitis, fewer and smaller tumors, and decreased β -catenin+ cells in the intestinal crypts³. Additionally, these mice exhibit lower levels of key cell cycle regulators, such as cyclin D1 and cyclin-dependent kinase 2, suggesting that IL-17A is crucial for both tumor initiation and progression. In our study, KD-fed mice displayed decreased Th17 cells and associated cytokines (Extended Figure 2f), suggesting that the ketogenic diet reduces this immune cell subtype, which may account for the observed reduction in tumor burden in KD-fed mice.

We had already mentioned in the submitted manuscript that we cannot conclude on whether the reduced immune cells are a result of the KD or of the lower number of tumors. We have now further elaborated this conclusion.

Updated text in lines 100-105 and 109-113:

Lines 100-105:

Characterization of full-length lamina propria (LP), mesenteric lymph node (MLN) and spleen immune cell profiles (Extended Data Fig. 2d and 2e) showed a reduction in LP CD4⁺ T cell populations in KD-fed mice. We particularly observed a decrease in Th17 cells (Extended Data Fig. 2f), a pathogenic subpopulation of CD4⁺ T cells known to play a critical role in CRC development as evidenced by reduced expression of IL-6, STAT3, IFN γ , and TNF α , leading to fewer and smaller tumors in IL-17A-deficient mice³.

Lines 109-113:

Thus, it remains unclear whether the observed reduction in immune cell populations is a result mediated by the KD or if it is a direct consequence of decreased tumor development. Conducting time-course analyses at various stages of the AOM/DSS-induced carcinogenic process would be essential to elucidate the temporal sequence of events leading to the observed reduction in colonic tumor burden.

New Extended Figure 2g and h

New Extended data figure 2. g-h. Expression of S100A8 in the colons of SD and KD mice in the GF (g) and SPF (h) dietary experiments, as detected by qPCR.

Line 103-105: The authors mention selecting a specific CRC donor rather than mixing stool samples from multiple donors due to high heterogeneity in microbial profiles. However, it is suggested that obtaining several samples from different donors and creating a mixture could reduce heterogeneity and provide a more representative microbiome for FMT. The authors didn't observe significant differences from mice that received FMT from a CRC donor and concluded that the microbiome of CRC patients did not exhibit a higher potential for CRC induction. However, if they had selected another CRC donor, it is likely to have yielded very different results, and the authors would have reached different conclusions. For this reason, the results from mice that received FMT from a single CRC donor seem inappropriate and nonreproducible for measuring the effects of the microbiome on CRC induction and should be removed from the manuscript.

We agree with the reviewer that basing our conclusions on a single CRC donor is insufficient. Since this experiment is not essential to the conclusions of the manuscript, we have decided to remove it from the revised manuscript. Text and Supplementary Figure 3 have been adapted accordingly.

Line 105-113: Overall, the results and conclusions derived from these results are contradictory and not persuasive. For instance, in the low-grade inflammatory model, no significant differences in colonic tumor burden and immune cell profiles are observed. However, the conclusion drawn is that "KD exerts an anti-tumorigenic effect in CRC, potentially by modulating the immune system." This raises questions about why the Ketogenic diet did not affect tumor burden and immune cell profiles in the inflammatory model and how such a conclusion was reached with non-significant results.

We understand the reviewer's concern. We try here to clarify our reasoning. There are observable differences in immune populations between KD and SD in the AOM/DSS model, which are not apparent in the lowly inflammatory model (AOM alone). This discrepancy could be attributed to the absence of damage-associated molecular patterns (DAMPs) or other signals that occur when DSS damages the epithelium. It is plausible that the immune system is activated following DSS-induced damage, and the KD somehow attenuates this response, leading to the observed difference in colonic tumor burden. Conversely, in the absence of high inflammation (i.e., no damage signals following DSS treatments), the KD does not further suppress immune responses below the normal baseline and thus the KD has no effect on tumor burden.

We have clarified the text in the manuscript in lines 119-123:

The differences in immune populations between the KD and SD in the AOM/DSS model, which are absent in this low-grade inflammatory model, may be due to the lack of damage-associated molecular patterns (DAMPs) from DSS-induced epithelial damage. This suggests that the KD attenuates chronic inflammation activated by such damage, potentially leading to a reduced tumor burden in this inflammation-driven CRC model.

Line 131: The authors tested the effect of palmitic/oleic acid treatment on Th17 differentiation. Whereas, stearic acid is considered the most important fatty acid in reducing tumorigenesis. It is questioned why the authors did not test this fatty acid in Th17 differentiation.

We thank the reviewer for this valuable comment. We have now addressed the missing link. We show in the updated Extended Data Fig. 8e the differentiation of Th17 cells upon stearic acid supplementation. We demonstrate that stearic acid supplementation reduces the

differentiation of CD4⁺ T cells into Th17 cells, a response which was not observed with other lipids.

Updated text in lines 247-249:

Furthermore, subsequent T cell differentiation assays demonstrated that stearic acid supplementation specifically inhibited differentiation of CD4⁺ T cells into Th17 cells, a response not observed with other fatty acids (Extended Data Fig. 8e and 8f).

New Extended data figure 8. e-f. Th17 differentiation upon treatment with BSA-complexed dodecanoic acid, oleic acid and stearic acid (50μM, e) and with BSA-complexed palmitic and oleic acid (10μM, f). Data is shown as mean±SD in (a), (b, right), (c), (d), (e) and (f). Data in (a), (c) and (c) shows n=8 mice per condition. Data in (e) shows n=4 and data in (f) shows n=6 biological replicates (mice) indicated by different datapoint shapes, from one experiment each.

Additionally, we have now performed experiments where we tested the direct effect of stearic acid in vivo. Alongside the in vitro results, we could show that stearic acid supplementation via dietary intervention reduces Th17 cells in vivo. These new data is now presented in Figure 6 and Extended Figure 8d.

New Figure 6. The microbial metabolite stearate exhibits anti-cancer effects. | **a.** Schematic representation of the dietary stearic acid supplementation experimental setup (created with BioRender.com). **b.** Compositions of the control diet (CD) and the stearate-supplemented diet (SSD), with % of calories from the indicated source. **c-d.** Stearic acid levels detected in serum (**c**) and fecal (**d**) samples from CD- and SSD-fed mice, as detected by LC-MS. **e.** Colonic tumor burden in CD and SSD mice. **f.** Representative images of colons from one CD and SSD mouse. Scale bar = 1mm. Data shows mean±SD from n=6 CD and n=5 SSD mice in (**c**), from n=8 CD and n=8 SSD mice in (**d**). Two-sided Mann-Whitney test in (**c**) and (**d**). Data shows mean±SD from n=8 CD and n=8 SSD mice in (**e**). Analysis of deviance of a negative-binomial generalized linear model.

New Extended data figure 8. d. Frequency of IL-17+ CD4+ in colonic lamina propria in CD and SSD mice, as detected by flow cytometry.

Updated lines 238-249:

Stearic acid reduces colonic tumor burden by inducing apoptosis in cancer cells

To further investigate the role of stearic acid in CRC development, we supplemented stearic acid through the dietary regimen to mice undergoing the same AOM/DSS CRC model described above (Fig. 6a, 6b and Extended Data Fig. 8a) and confirmed increased free stearic acid levels in the stearate-supplemented diet compared to the control diet (Extended Data Fig. 8b). Mice on SSD also exhibited elevated stearic acid serum and fecal levels (Fig. 6c and d, Supp. Tables 7 and 8). We observed a reduction in colonic tumor burden (Fig. 6e and 6f), indicating that stearic acid exerts an anti-tumorigenic effect in the development of CRC. We did not observe any difference in fecal lipocalin-2 levels between groups (Extended Data Fig. 8c), however, we further linked the reduced tumor burden to decreased Th17 cells in vivo (Extended Data Fig. 8d). Furthermore, subsequent T cell differentiation assays demonstrated that stearic acid supplementation specifically inhibited differentiation of CD4⁺ T cells into Th17 cells, a response not observed with other lipids (Extended Data Fig. 8e and 8f).

Line 141-144: It is unintelligible why the authors set this study design (utilizing open-top cages in a gnotobiotic flexible-film isolator to allow for microbial exchange between groups) to confirm the mediating effect of the microbiome on the anti-cancer effects of KD. For this type of research question, fully separate housing to inhibit microbial exchange between groups would be better to observe the modulation effect of the gut microbiome. As such, it remains questionable whether mice having different microbiome profiles will lead to different results in anti-cancer effects.

We thank the reviewer for this comment. Our results suggest that the ketogenic diet exerts a strong lasting selective pressure on the gut microbiome (previous Figure 2). However, this specific experiment indicates that the diet is unable to maintain selective pressure when microbial exchange is possible, even with continuous dietary adherence. It is unclear why the KD maintains only a “weak” selective pressure under these conditions. We value this experiment as it shows that ketosis alone is insufficient to ensure a reduction in colonic tumor burden, which is still debated in the field. Nevertheless, we agree that the CMT experiment provides a much more powerful setup to conclude on the causal role of the microbiome in the anti-tumorigenic effects of the KD. Therefore, we have decided to remove this experiment from the updated manuscript.

Additionally, according to the study design figures, it seems the authors performed FMT several times throughout several weeks. However, FMT is ideally or mostly conducted at the very early stages of experiments for microbiome transplantation to observe the long-lasting effect of FMT.

We thank the reviewer for this valuable comment. The mouse microbiome cannot be fully eradicated in an SPF model/facility, thus constant enrichment of the donor microbiome is needed, in order to draw conclusions relevant to the human microbiome, rather than the resident mouse microbiome. FMT was ceased when mice were switched to the ketogenic diet, allowing the diet to exert selective pressure over time. In the CMT experiment, we decided to gavage the cecal content of the donor mice throughout the experiment to maintain a resident microbiome as close to the donor microbiome as possible, as the goal was not to assess changes occurring over time when a ketogenic-induced microbiome is exposed to standard dietary conditions, but rather to evaluate the functional role of those changes which have already occurred under the selective pressure of the ketogenic diet in the donor mice.

Updated text in lines 91-94:

Antibiotic-treated SPF mice were subjected to bi-weekly FMT during the induction of CRC to maintain enrichment of the donor fecal microbiome in relation to the residual host mouse microbiome. FMT was ceased at diet change to allow for a natural evolution of the gut microbiome in response to the diet and CRC progression.

Line 181: Various *Alistipes* spp. were differentially abundant, but it is unclear whether this bacterium is associated with the anti-cancer effect throughout the manuscript. The author should evaluate and discuss the implications of the enrichment of this bacteria and whether it contributes to the anti-cancer effect.

We thank the reviewer for this comment and have now added more information on *Alistipes* spp. The genus *Alistipes* has only recently been thoroughly described, and members have since been identified in human and mouse microbiome samples. Due to their prevalence and abundance in the gut, *Alistipes* species have been correlated with different disease parameters and potentially play roles in different disease contexts, including CRC⁴. In light of recent reports, it becomes evident that *Alistipes* cannot be considered strictly pathogenic or beneficial; different species appear to play distinct roles in different disease contexts. Therefore, we performed a more detailed investigation of the genome content of each species/MAG to identify *Alistipes* species and observed specific species and strains enriched in KD-fed mice (Figure 1 for reviewing purposes). Nevertheless, we will follow-up on the role of *Alistipes* in the KD-mediated anti-tumorigenic effect in CRC.

Figure 1 for reviewing purposes: Pan genome analysis of *Alistipes* sp002428825. MAGs (metagenomics-assembled genomes) of the *Alistipes* species sp002428825 assembled from reads of feces samples collected from SC and KC mice were compared using the tool anvio revealing three gene cluster specific for the SC- (Std) or KC- (Keto1/2) derived MAGs suggesting a selection of certain strains by ketogenic diet or in-host evolution.

Updated text in lines 184-186:

However, the genus *Alistipes* was only recently described, and although the genus has since then been linked to various diseases, including colitis and CRC, the role of *Alistipes* in disease is still unclear⁴.

Given the considerable differences in the microbial profiles of CMT-donor mice and recipient mice, significant differences in their functional profiles are already expected. Therefore, the description of the functional profiles should focus on finding consensus functional profiles between CMT-donor and recipients, despite significant differences in profile, rather than describing it as if it ‘maintained’.

We thank the reviewer for this comment. A detailed analysis of the functional profiles of the microbiome from SD/KD and SC/KC mice revealed a number of persistently different functional features. Most strikingly, we identified an expected shift in the CAZyme (carbohydrate active enzymes) profiles between mice (Figure 2A for reviewing purposes). Briefly, the microbiome of mice on KD partly lost their capacity to degrade dietary sugars, while CAZymes associated with mucus degradation became more abundant. This is probably a direct response to the lack of dietary sugars in the ketogenic diet. Moreover, we identified a higher abundance of genes associated with flagellar synthesis and chemotaxis, known to be particularly encoded by Firmicutes species, such as *Clostridia*, in KD and KC mice than the standard diet groups (Figure 2B for reviewing purposes). In addition, we more recently identified more subtle changes in the functional profiles related to energy metabolism, but also to vitamin and cofactor synthesis (Figure 2C-F for reviewing purposes). However, due to the complexity of the functional profiles, future work will aim to identify causality between

the enrichment/depletion of functional features. We have now removed data and statements related to functional profiles of these microbiomes and intend to publish our findings in another article, in more detail.

Figure 2 for reviewing purposes: For the analyses of the functional capacity of microbiota from different groups, open-reading frames predicted using Prokka were annotated with a hidden Markov models (HMM) approach using a database including KOs derived from the Kyoto Encyclopedia of Genes and Genomes (KEGG) and CAZyme databases. (A+B) Volcano plots showing CAZymes identified in increased and reduced abundance in microbiomes from (A) KD- vs SD-fed mice and in microbiomes from (B) KC vs SC mice. (C-F) KOs from different functional pathways were consistently more or less abundant in mice from ketogenic vs standard conditions. Shown are clr-transformed read counts.

Updated text in lines 204-235:

KD induces a shift toward stearic acid production, which is maintained upon FMT

To assess whether and how the microbiome contributes to shaping the LCFA pool, specifically stearic acid, we focused on microbial LCFA metabolism. Most bacteria can produce LCFAs, which are essential bacterial membrane components and signaling molecules. LCFAs are produced in a multistep process by enzymes that are highly conserved among bacteria (Fig.

5a). Nevertheless, membrane LCFA profiles differ between microbial taxa, varying for instance in chain length and degree of saturation⁵. To investigate how the shift in the microbial communities reported in Fig. 3 affected the intestinal LCFA pool in KD-fed and KC mice, we first mapped our metagenomic sequences against the Kyoto Encyclopedia of Genes and Genomes (KEGG) database to obtain a list of contigs carrying LCFA metabolism-associated KEGG orthologs. These contigs were further annotated at the taxonomic level using Kraken2 to estimate the contribution of different microbial classes to LCFA metabolism in the different dietary conditions. Consistent with the taxonomic shift reported in Fig. 3i, the relative contribution of bacteria from the class Clostridia was higher in the KD groups, while members of the Bacteroides were diminished resulting in less LCFA metabolism-associated contigs assigned to this class (Extended Data Fig. 7b). Specifically, contigs carrying the microbial acyl-CoA thioesterases TesA and FatA (Fig. 5b) — enzymes required for the release of free LCFAs, such as stearic acid, from the acyl-carrier⁶ — were more frequently assigned to species from the class Clostridia in the KD groups compared to the SD groups. This suggests that in the KD groups, members of the class Clostridia may have contributed more significantly to the intestinal free LCFA pool (Extended Data Fig. 7b). To investigate whether stearic acid is produced by Clostridia, we analyzed a dataset published by Han et al.⁷, who recently profiled the metabolome of bacterial culture supernatants. Visualization of log₂ fold changes in stearic acid levels in these supernatants revealed that Bacteroidaceae and Tannerellaceae — two families within the class Bacteroidia — predominantly consume stearic acid in vitro. In contrast, families from the class Clostridia, such as Lachnospiraceae and Ruminococcaceae, release stearic acid into the culture supernatants (Fig. 5c). Notably, these families, although not necessarily predominant in the communities, were consistently enriched or depleted across experiments in KD and KC mice, respectively (Fig. 5d). Furthermore, the analysis of two other abundant LCFAs, palmitic and oleic acid, showed that palmitic acid was consumed by most bacteria, with the exception of some Clostridiaceae, while oleic acid was released by all bacteria. This finding suggest that stearic acid is a family-specific LCFA among the three analyzed LCFAs (Extended Data Fig. 7c).

Line 333-335: The authors mention the difference between responder and non-responder, which is based on their different microbial profiles. However, since all the mice in this study received FMT, and the basic assumption of FMT is that the recipient will share a similar microbial profile with the donor and other mice receiving the same FMT, the purpose and assessment of microbiome effects on health are unclear. If they indeed had different microbial profiles after FMT, then what is the purpose of FMT, and how can the microbiome effects on health be assessed?

We thank the reviewer for this comment. Mice do not have different profiles after FMT, on the contrary, their profiles are very similar, and it is the selective pressure of the ketogenic diet which induces changes in the gut microbiome and leads to a reduced colonic tumor burden. They are not however precisely identical, small differences do exist between mice

after FMT, and therefore between mice after ketogenic diet consumption. The concept of “responder” and “non-responder” is however drawn from the regavaging experiment and from the fact that there is a large variability in the number of colonic tumors in the CMT experiment (Figure 2b), suggesting that some of the microbiomes collected from the ketogenic diet experiment for FMT were more “potent” than others. This notion of “responder” and “non responder” has also already been proposed in the context of epilepsy, where KD is regularly used for the reduction of seizures.

We have now clarified the text in the updated lines 272-279:

The gut microbiome plays a key role in mediating the anti-cancer effects of the KD in the context of a murine model of inflammation-driven CRC. However, the variability in the response observed (Figure 2b) suggests that the initial microbiome profile may determine whether an individual is a "responder" or "non-responder" to KD. This concept, explored in epilepsy patients undergoing KD therapy for the reduction of seizures and in cancer treatments like immune checkpoint inhibition, highlights the importance of the gut microbiome. Future studies should focus on identifying biomarkers to predict response and strategies to convert "non-responders" into "responders," paving the way for personalized treatments.

Methods: 1) The methods for extracting genomic DNA from fecal samples for sequencing and preparing stool/plasma samples for metabolomic profiling are omitted.

Detailed information on DNA extraction and preparation of stool samples for metabolomics profiling can be found in the methods section, and was already included in the original manuscript.

2) The specific 16S region used for experimentation in the '16S rRNA sequencing data analysis' section is not mentioned.

We used the V4 region of 16S rRNA genes using primers 515F and 805R, and primer sequences can be found in the corresponding supplementary table, also included in the original manuscript.

3) Also, DADA2 generates ASVs not OTU.

Thanks for pointing out this error, which has been corrected in the revised manuscript.

4) The correct name of the taxonomy database name is ‘SILVA’.

This has been corrected in the revised study.

5) There are profound differences between SILVA versions 132 and 138. Why did the authors utilize both versions instead of just the current/latest one?

We thank the reviewer for pointing this out. We initially used SILVA version 132 for 16S data analysis and the PCA plots displayed in Figures 3a and b are based on this analysis. Changes of community profiles over time, however, have been re-analyzed at a later stage of the project to ensure comparability between the analysis workflow used for GF and SPF mice. At this point of time, we used SILVA version 138. We have clarified the use of both SILVA versions in the methods section.

Figure 5d: All the ketogenic mice have either no tumors or very few (seemingly 1-2), making it doubtful whether the ketogenic diet truly has such a strong effect in inhibiting tumorigenesis.

We thank the reviewer for this comment. We confirm that our observations are accurate. All of our studies are carefully controlled, and tumor numbers are blindly scored by two independent researchers. We have now included more representative images to better illustrate the tumors we observed in all experiments. Additionally, we have performed three experiments in two different models (one GF and two SPF experiments) to address our research questions, and both models show the same findings.

Ext. Figure 4e-f: The authors provided differences in BAs but did not describe the significance of these differences. It is better to provide implications rather than just showing differences.

We agree with the reviewer. To increase the robustness of our findings, we consistently based our next steps on observations from both the GF and SPF experiments. Since these two experiments showed opposing trends regarding bile acids, we decided not to pursue this line of investigation further, however we include these negative findings in the extended data figures to demonstrate that this was an avenue we explored and excluded.

Ext. Figure 6: The group labels 'Standard GF' and 'Ketogenic GF' do not match with the group labels used in the Results section (SD-CMT and KD-CMT). These labels may lead to misinterpretation as they indicate germ-free status. It is recommended to change the labels to match those used in the Results.

The differences in labeling across figures have been corrected.

Reviewer #2

In this manuscript, Tsenkova and colleagues find that stearate, a long-chain fatty acid that is produced after ketogenic diet feeding, can reduce colorectal cancer burden. The authors describe, in great detail, the phenotypic, metabolic, and gut bacterial/immune composition changes that occur in mouse models under ketogenic diet. This is an extensive study describing how the ketogenic diet might influence responses to CRC, and the findings presented in this paper can fill the gap between KD-altered gut microbiota and CRC development. However, additional evidence is needed to support the conclusions drawn by the authors.

Major comments

1. The manuscript starts by detailing many correlations with KD and gut/immune phenotype, but do not connect these final observations of stearate upregulation with initial observations, keeping in mind that a major conclusion of the paper is that the KD-derived stearate reduces CRC tumor burden, this should be connected.

We thank the reviewer for this comment. We have now further connected the phenotype to stearic acid. We performed an experiment under GF conditions where mice with a humanized gut microbiome were subjected to an AOM/DSS model and fed stearic acid-rich diet (New Fig 6A, 6B, Extended Data Fig. 8A and B). These mice exhibited increased levels of stearate in their plasma and feces, as compared to those consuming the control diet (new Figure 6C and D). At the endpoint, we observed a reduced colonic tumor burden in mice fed the stearate-supplemented diet (new Figure 6E), accompanied by reduction in Th17 cells (new Extended Data Figure 8D). Furthermore, we confirmed these findings by performing Th17 differentiation assays, in which we observed that the differentiation of Th17 cells was impaired by stearic acid, but not by other long-chain fatty acids (new Extended Figure 8E and 8F). Altogether, these results are consistent with our in vivo findings and further strengthen the link between KD-induced gut microbiome profiles and the effects of stearic acid on colorectal cancer.

Updated text in lines 239-249:

To further investigate the role of stearic acid in CRC development, we supplemented stearic acid through the dietary regimen to mice undergoing the same AOM/DSS CRC model described above (Fig. 6a, 6b and Extended Data Fig. 8a) and confirmed increased free stearic acid levels in the stearate-supplemented diet compared to the control diet (Extended Data Fig. 8b). Mice on SSD also exhibited elevated stearic acid serum and fecal levels (Fig. 6c and d, Supp. Tables 7 and 8). We observed a reduction in colonic tumor burden (Fig. 6e and 6f), indicating that stearic acid exerts an anti-tumorigenic effect in the development of CRC. We did not observe any difference in fecal lipocalin-2 levels between groups (Extended Data Fig. 8c), however, we further linked the reduced tumor burden to decreased Th17 cells in vivo (Extended Data Fig. 8d). Furthermore, subsequent T cell differentiation assays demonstrated

that stearic acid supplementation specifically inhibited differentiation of CD4⁺ T cells into Th17 cells, a response not observed with other lipids (Extended Data Fig. 8e and 8f).

New Figure 6 and Extended Figure 8:

New Figure 6. The microbial metabolite stearate exhibits anti-cancer effects. | a. Schematic representation of the dietary stearic acid supplementation experimental setup (created with BioRender.com). **b.** Compositions of the control diet (CD) and the stearate-supplemented diet (SSD), with % of calories from the indicated source. **c-d.** Stearic acid levels detected in serum (**c**) and fecal (**d**) samples from CD- and SSD-fed mice, as detected by LC-MS. **e.** Colonic tumor burden in CD and SSD mice. **f.** Representative images of colons from one CD and SSD mouse. Scale bar = 1mm. Data shows mean±SD from n=6 CD and n=5 SSD mice in (**c**), from n=8 CD and n=8 SSD mice in (**d**). Two-sided Mann-Whitney test in (**c**) and (**d**). Data shows mean±SD from n=8 CD and n=8 SSD mice in (**e**). Analysis of deviance of a negative-binomial generalized linear model.

New Extended data figure 8. | a. Mouse body weight (grams) over time (days) in stearate-supplemented diet-fed (SSD) mice and control-diet-fed (CD) mice. **b.** Free long-chain fatty acid profiles (**left**, heatmap shows means) and free stearic acid content detected in SD and KD (**right**, shown as mean±SD). Two technical replicates per diet are shown. **c.** Fecal lipocalin-2 levels (pg per gram of feces) in CD and SSD mice as detected by ELISA. **d.** Frequency of IL-17⁺ CD4⁺ in colonic lamina propria in CD and SSD mice, as detected by flow cytometry. **e-f.** Th17 differentiation upon treatment with BSA-complexed dodecanoic acid, oleic acid and stearic acid (50μM, **e**) and with BSA-complexed palmitic and oleic acid (10μM, **f**). Data is shown as mean±SD in (**a**), (**b**, **right**), (**c**), (**d**), (**e**) and (**f**). Data in (**a**), (**c**) and (**d**) shows n=8 mice per condition. Data in (**e**) shows n=4 and data in (**f**) shows n=6 biological replicates (mice) indicated by different datapoint shapes, from one experiment each. Unpaired Mann-Whitney U test in (**c**). Unpaired T test in (**d**), data passed Shapiro-Wilk normality test p = 0.29. Paired ordinary one-way ANOVA in (**e**) and (**f**).

2. Why would reduced populations of immune infiltrates (ED Fig 2) cause or be a result of reduced tumor burden? Is this reduced tumor burden merely reflective of the increased stearate in the circulation from KD that is found to suppress CRC?

The AOM/DSS model mimics an inflammation-driven process, representing CRC patients with pre-existing inflammation. In this model, AOM is the carcinogen which induces mutations in

the epithelial cells of the colon. The DSS causes damage to the colonic epithelium, leading to inflammation and ulceration. This inflammatory environment, mostly driven by IL17-producing cells⁸, promotes the progression of AOM-induced pre-cancerous lesions to malignant tumors. We would like to clarify that we administer the ketogenic diet after the last cycle of DSS is completed, mimicking a therapeutic setting and not a prevention model. Furthermore, we performed lamina propria immune cell isolation on whole colons at endpoint, thus we can only comment on overall colonic immune populations, not on tumor-specific immune profiles between groups. Nevertheless, our data align with published literature on KD, which shows that KD reduces Th17 cell abundance in the gut⁹.

As for the second question raised by the reviewer, we want to clarify that we find the stearic acid to be increased in the fecal samples of the KD-fed mice and not in the circulatory system, suggesting that it is not a systemic effect of the KD but rather that it is the KD-driven changes in the gut microbiome that lead to differences in stearic acid. We have now further strengthened the link between KD and stearic acid by demonstrating that the difference in free stearate levels in the gut lumen of KD-fed mice does not originate from the diet itself (new Extended Data Fig. 7D), and by further performing additional in vitro and in vivo experiments showing that direct supplementation of stearic acid reduces tumor burden in vitro and in vivo. (new Figure 6). We could also confirm that stearic acid has a direct impact on Th17 differentiation (new Extended Figure 8E).

New Extended data figure 7. d. Free long-chain fatty acid profiles (**left**, heatmap shows means) and free stearic acid content detected in SD and KD (**right**, shown as mean±SD). Two technical replicates per diet are shown.

3. Does oral administration of stearate reduce CRC burden in the AOM/DSS model?

Please refer to response to comment number 1 from the same reviewer and to the new Figure 6 and Extended Figure 8.

4. Th17 downregulation is noted in the mouse models. What is the role of Th17 CD4+ T cells in the control of CRC tumors? If their reduction/downregulation is important for KD-

mediated tumor control, is there a benefit in tumor control from using CD4-T cell depletion in the AOM/DSS CRC model in SD-mice?

In the AOM/DSS model, cancer progression is driven by immune cells, including IL-17-producing cells⁸. Th17 cells, in particular, are clinically relevant in CRC; elevated levels of IL-17A and ROR γ t in CRC patients' tumor tissues are associated with a significant reduction in disease-free survival, highlighting the pro-tumorigenic role of Th17 cytokines^{1,2}. This is further supported by evidence from IL-17A-deficient mice in the AOM/DSS model, which show reduced expression of IL-6, STAT3, IFN γ , and TNF α , leading to milder colitis, fewer and smaller tumors, and decreased β -catenin⁺ cells in the intestinal crypts³. Additionally, these mice exhibit lower levels of key cell-cycle regulators, such as cyclin D1 and cyclin-dependent kinase 2, suggesting that IL-17A is crucial for both tumor initiation and progression. Thus, as questioned by the reviewer, the specific targeting of Th17 cells in the AOM/DSS model has already been demonstrated to be beneficial. In our study, KD-fed mice displayed decreased Th17 cells and associated cytokines, suggesting that the ketogenic diet reduces this immune cell subtype, which may account for the observed reduction in tumor burden in KD-fed mice. Nevertheless, we have also now additionally performed RNA sequencing of stearic acid stimulated CRC cell lines and identified a direct pro-apoptotic effect on cancer cells.

Updated text in lines 254-262:

We then performed RNA sequencing on HT-29 CRC cells treated with stearic, oleic and dodecanoic acids to investigate the molecular mechanisms underlying the observed anti-cancer effect. Gene set enrichment analysis (GSEA) suggested an arrest of cell proliferation and the induction of apoptosis upon stearic acid treatment (Fig. 6k and 6l). Furthermore, stearic acid treatment was able to revert cell cycle and apoptosis related gene expressions observed in control (vector-treated) cancer cells, which are in line with those observed in patient tumor samples compared to matching normal tissue samples in the TCGA dataset (Fig. 6l, Extended Data Fig. 8k). The use of a broad caspase inhibitor was able to abrogate the effects of stearic acid treatment, confirming the underlying mechanism of stearic acid-induced cancer cell death (Fig. 6m and n).

New Figure 6. The microbial metabolite stearate exhibits anti-cancer effects. Principal component analysis of HT-29 cells treated with 200 μM of oleic, dodecanoic and stearic acid. **k.** Hallmark gene sets in stearic acid-treated versus control cells. **l.** Cell cycle and apoptosis related gene expression in control, oleic, dodecanoic and stearic acid-stimulated HT-29 cells and in tumor and matching normal tissue samples. **m-n.** Caspase-3 activity in HT-29 (**m**) and HCT-116 (**n**) cells treated with stearic acid in presence or absence of a broad caspase inhibitor (zVad). Staurosporine 2uM was used as a positive control. Data shows mean±SD of n=3 independent experiments (indicated by different datapoint shapes), in (**m**) and (**n**).

5. On this same note, it was mentioned (lines 125-127) that KD-derived ketone bodies - acetone and Na-acetoacetate - reduced Th17 differentiation. How does stearate regulate Th17 differentiation and in general, immune composition in these mice?

This experiment has now been performed. We indeed observed a reduced Th17 differentiation in the presence of stearic acid as well as a reduced Th17 immune cell population in mice fed with a stearic acid enriched diet (new Extended Data Fig. 8E, D and F).

Updated text in lines 246-249:

We further linked the reduced tumor burden to decreased Th17 cells in vivo (Extended Data Fig. 8d). Furthermore, subsequent T cell differentiation assays demonstrated that stearic acid

supplementation specifically inhibited differentiation of CD4⁺ T cells into Th17 cells, a response not observed with other lipids (Extended Data Fig. 8e and 8f).

New Extended data figure 8. d. Frequency of IL-17⁺ CD4⁺ in colonic lamina propria in CD and SSD mice, as detected by flow cytometry. e-f. Th17 differentiation upon treatment with BSA-complexed dodecanoic acid, oleic acid and stearic acid (50µM, e) and with BSA-complexed palmitic and oleic acid (10µM, f). Data is shown as mean±SD in (a), (b, right), (c), (d), (e) and (f). Data in (a), (c) and (c) shows n=8 mice per condition. Data in (e) shows n=4 and data in (f) shows n=6 biological replicates (mice) indicated by different datapoint shapes, from one experiment each.

6. Following this point, does KD-CMT induce Th17 reduction, or general immune cell reduction, as shown in the KD-mice of ED Fig 2d/e? Clearly, this is an important connection to link the role of the microbiome with any speculation of immune-mediated tumor control.

We thank the reviewer for this comment. Unfortunately, during the CMT experiment, the FACS machine broke down, preventing us from assessing Th17 cells in the colon during the CMT experiment. However, we have now measured the levels of IL17 in cryopreserved colons from CTM mice by qPCR, serving as a proxy for Th17 cells. We observed a reduction in IL17 levels, consistent with our previous in vivo findings in KD-fed mice.

Updated text in lines 159-161:

Furthermore, colonic IL17 expression levels were decreased in KC mice compared to SC mice (Extended Data Fig. 6e), in line with the decreased number of Th17 cells in the KD-fed mice (Extended Data Fig. 2f).

New Extended Figure 6. IL-17 (e) gene expression in KC and SC mice in the CMT experiment, as measured by qPCR. Data is shown as mean±SD. Two-tailed Mann-Whitney U test.

7. The authors tried to assess KD suppress CRC via modulating inflammation by using AOM model instead of AOM/DSS. However, in addition to being an inflammatory agent, DSS treatment also compromises epithelial barrier integrity, changes microbial composition, which allowing the bacteria and bacteria products invade into underlying tissue. In combination of suggestion 6 and 11, the results are not robust enough to state the anti-tumorigenic effect of KD in CRC is mediated by modulating the immune system (line 112).

We have now further analysed barrier integrity in KD fed mice. We did not observe a difference in occludin expression, neither in LPS levels in the plasma, nor in colon mucus layer thickness, findings which are aligned with existing literature on KD. Indeed, studies of KD in mice have shown that mucus layer thickness does not change upon KD, despite the lack of fermentable carbohydrates and the KD-driven gut microbial shift is independent of colonic mucin¹⁰. Furthermore, occludin expression was also unaltered in KC recipients when compared to SC recipients in the CMT experiment, indicating that the KD-induced microbial changes were not accompanied by an increased capacity for mucosal layer erosion and bacterial translocation (new Extended Figure 5e and f and New Extended Figure 6b and 6c).

Updated text in lines 149-156:

We first excluded bacterial translocation as a primary factor in the KD-mediated phenotype, as mucus layer thickness (Extended Data Fig. 5a and 5b (GF)), LPS levels in plasma (Extended Data Fig. 5c (GF) and 5d (SPF)), and occludin expression (Extended

Data Fig. 5e and 5f (SPF) remained unaltered upon KD consumption, in line with Ang et al.⁹ We next performed cecal microbial transplantation (CMT) of samples collected from the SPF diet mice (Fig. 1d) into a next generation of SD-fed GF mice undergoing the same cancer model (Fig. 2a, Extended Data Fig. 6a). We did not observe a statistically significant difference in colonic occludin expression in ketogenic cecal recipient mice (KC, Extended Data Fig. 6b and 6c) when compared to standard cecal recipients (SC).

New Extended Figure 5. e-f. Colonic occludin gene expression (**e**) and protein expression (**f**) in KD and SD mice in the SPF dietary experiment, as measured by qPCR (**e**) and immunofluorescent staining (occludin-TRITC in red and DAPI nuclear stain in blue, one representative colon per condition is shown, scale bar = 200 μ M, **f**). Two-tailed Mann-Whitney U test in (**a**), (**c**), (**d**), (**e**) and (**g**). Data in (**a**) shows n=4 SD and n=6 KD mice. Data in (**c**) and (**d**) shows n=4 mice per condition. Data in (**e**) shows n=6 mice per condition.

New Extended Figure 6. b-c. Colonic occludin gene expression (**b**) and protein expression (**c**) in KC and SC mice, as measured by qPCR (**b**) and immunofluorescent staining (occludin-TRITC in red and DAPI nuclear stain in blue, one representative colon per condition is shown, scale bar = 200 μ M, **c**).

Minor comments:

1. No figure legends.

We apologize for this significant omission, which was due to a copy/paste issue on our side. We have now ensured that the figure legends have been added.

2. In Line 96, the authors state calprotectin level was reduced in the spleen of GF KD-fed mice. Calprotectin is a common diagnostic evaluation of IBD patients, but usually, the fecal calprotectin is used. How's the fecal calprotectin level in this study? what is the rational for using spleen calprotectin level to indicate gut inflammation?

We agree with the reviewer and have now performed qPCR of S1008A in the colon. We do not observe a difference in the expression of these inflammation markers, suggesting that the overall inflammatory milieu of the gut lumen is similar between the conditions.

Updated text in lines 106-109:

The expression of S1008A, a subunit of calprotectin (a biomarker of inflammation used in inflammatory bowel disease¹¹), was however unchanged in GF and SPF KD-fed mice, suggesting an overall similar inflammatory milieu (Extended Data Fig. 2g,h).

New Extended data figure 2. g-h. Expression of S1008A in the colons of SD and KD mice in the GF (g) and SPF (h) dietary experiments, as detected by qPCR.

3. The schema of Figure 1a is not clear, for example, there appears to be overlap between antibiotic treatment and 5HD gavage.

Yes, this is correct. 5HD gavage commences at the moment that antibiotics are removed. Therefore, there is an effective overlap in the scheme, as these two events occur on the same

day. The timelines of all experiments are also clearly indicated in the corresponding materials and methods sections.

4. Line 211, in this part, the authors identified that the microbial metabolite stearate is elevated by KD and exhibits anticancer effects. This is a very interesting finding, and it is also the major conclusion of this paper. It is too long to be one section. It would be better to separate this section into three parts: 1. Fecal metabolic profile analysis indicates an increase in stearic acid level in KD-fed mice. 2. Elevated stearic acid level is associated with changes in bacterial taxa that related to fatty acids producing and/or consuming. 3. In vitro treatment or in vivo supplementary of stearic acid can suppress CRC progression.

We agree with the reviewer and have now structured the part on the stearic acid differently and included the new in vitro and in vivo data in a new separate figure as suggested by the Reviewer. Please refer to new Figure 6 and to previous comments by the same Reviewer.

5. Line 102 What is the rationale for using CRC patient FMT in AOM model if it's not mentioned in the AOM/DSS model? Would there be any difference between CRC-FMT and 5HD-FMT in AOM/DSS model?

We agree with the reviewer that this data is not strong enough to support the absence of difference between CRC-FMT and 5HD-FMT. In agreement with a comment raised by Reviewer 1, we have removed this experiment, which involved a single gavage of one CRC donor, from the manuscript.

6. Is the low-inflammatory model in ED Fig 3 capable of generating sufficient colonic tumors for analysis? i.e., is it possible that we do not see statistical differences in SD vs KD simply because the model is not capable of generating sufficient colonic tumors?

We have established the single AOM model in our lab some years ago and we use this model frequently. In our published paper¹², we were able to see differences in the AOM model, despite an overall lower number of colonic tumors compared to the AOM/DSS model. A comparable number of colon tumors has also been observed in the APC min model, with its use resulting in notable differences across various studies^{13,14}. Therefore, the number of tumors we observe is consistent with existing literature, allowing us to draw meaningful conclusions from the findings.

7. Please also show the antibiotic depletion efficiency if you collected the stool after antibiotic treatment and before 5HD gavage. Additionally, what is the implantation efficiency? It would be beneficial to compare the composition of bacteria in the input 5HD oral gavage samples and stool samples after CMT.

To start with, we would like to mention that we have already successfully used the same antibiotic cocktail in our previous studies (de Nies, et al. 2022¹⁵, please refer to Figure 1 of the article to see the successful depletion of the murine microbiome upon the same antibiotic cocktail supplementation). The use of an antibiotic cocktail instead of single antibiotics results in broad-spectrum activity depleting most bacterial species present in the intestine leading to a significant depletion of the murine microbiome.

a

b

Figure 3 for reviewing purposes: Figure 1 (Experimental design and metagenome-assembled genome profiles) from de Nies et al (2022)¹⁵ demonstrating the depletion of most members of the microbiome following treatment with an antibiotic cocktail.

Since this procedure is well-established, we did not count viable bacteria per gram feces after antibiotic depletion. However, we observed a significant lower DNA content during fecal DNA extraction and lower file size of 16S gene sequencing files from samples collected right after antibiotic treatment compared to the follow up timepoints despite

the facts (1) that DNA was extracted from equal fecal biomass, (2) that equal amounts of DNA (5 ng) were provided as input for the 16S gene sequencing and (3) that all samples were sequenced at the same time. We are aware that neither DNA concentrations nor file size are an adequate measure to draw such a conclusion. However, in this case, we think that the lower DNA amounts and bacterial reads reflect the expected depletion of the microbiome following broad-spectrum antibiotic treatment (Figure 4 for reviewing purposes).

Figure 4 for reviewing purposes: (A) DNA concentration of DNA samples extracted from fecal pellets collected from mice after antibiotic treatment (after Abx), after gavage with the 5HD mix (after gavage), before and after diet change (before/after diet change), in the middle of the experiment (mid diet) and at its end (end diet). Each dot presents a DNA sample from an individual mouse. (B) 5 ng of DNA was subjected to 16S rRNA gene sequencing. Sizes of each individual sequencing file (in mb) were plotted for each mouse microbiome analyzed at the selected time points. Following diet change, field sizes are displayed for both diets separately to demonstrate that diet had no influence on file size.

In addition, we observed that microbiota of mice sequenced at different timepoints following gavage and before diet change were quite similar to the 5HD mix, comparable between most mice and stable over time. The stability becomes particularly obvious when we compare the microbiota from the same mice at two different timepoints between Abx+gavage and diet change.

Figure 5 for reviewing purposes: Barplots showing microbiome profiles on family level of the 5HD mix and of selected mice after gavage with the 5HD mix and directly before diet change. Relative abundance of

families is shown for the microbiota of individual mice (see strip label) on the selected days (displayed on x axis).

8. It would be better to have a SPF-stool-FMT control or germ-free stool-FMT control for GF group and SPF group, respectively.

We are not sure we understand the reviewer's comment. We are not interested in gavaging GF mice with stool from SPF mice, to compare to GF mice gavaged with stool from humanized SPF mice. There is also no purpose served by gavaging GF stool to GF mice, apart from maintaining a contamination control. There is no relevant comparison between the FMT experimental groups and GF mice given GF stool. Furthermore, GF die from DSS administered at the dose needed for the FMT experiment.

9. Line 93 and extended Fig. d and e are not clearly labeled and described. Were the samples from GF or SPF mice? what methods were used to quantify the immune cell profiles? CD4+ and CD8+ T cell populations, which are two major anti-tumor immune cells, were decreased in KD-fed mice. Could you explain the possibility?

Thank you for pointing this out. The (h) and (i) in the legend have now been changed to (d) and (e) respectively. The methods and gating strategies are described in the paper in the appropriate sections (Extended Data Fig. 9 and 10).

10. Line 103, the authors mentioned a single CRC donor was used for colonization because of dissimilarities of microbial profiles. What is the principle to choose this one CRC donor rather than others to perform FMT in this study?

Several studies suggest that each CRC donor has a unique microbiota. Therefore, mixing multiple fecal samples from CRC patients to generate an FMT is usually not recommended, as competition between different microbiomes might bias the outcome. However, we agree with the reviewer that additional FMTs with different CRC donors in GF mice are necessary to determine if FMT from CRC patients induces a different phenotype compared to healthy donor FMT. Consequently, we have decided to remove the CRC FMT data from Extended Figure 3, as it is not directly linked to the main message of the present manuscript.

11. Does the // in Extended Data Figure 3a means a discontinued diet treatment? If so, please explain.

Thank you for raising this concern. It is not a discontinued diet; it indicates the passing of several weeks on the diet. We have clearly indicated that the diet was maintained throughout the entire duration of the experiment (16 weeks) in the corresponding materials and methods section.

12. In Line 107, according to extended Data Fig. 3f, there are differences in immune cell profiles, eg. MLN macrophage population is different among GF, 5HD and CRC.

We thank the reviewer for this comment. Slight differences are observed but statistical significance is not reached for these comparisons.

13. No statistical analysis of Extended Fig. 4h. 10 mM, 50 mM and 100 mM of BHB are high doses for in vitro treatment. Could the lack of effect of BHB be due to the high concentration? What was the viability of the cells?

We thank the reviewer for this comment. The statistical analysis is described in the figure legend and no significant differences were observed. There was also no difference in viability between groups in these experiments. We provide here below viability graphs for reviewing purposes.

Figure 6 for reviewing purposes: Viability of T cells upon stimulation with different fatty acids.

14. Line 116, this whole section – KD induces a distinct host metabolic profile, is not associated with other parts in this paper. It is known that KD can alter metabolic profile.

This is an avenue we explored due to the literature on bile acids. However, we did not pursue this due to inconsistent findings as shown in Extended Figure 4e and f. Nevertheless, we consider it important to make this data available to our readers, and to highlight that this was an avenue we investigated and excluded.

15. Mucus layer thickness (ED Fig 5a) is clearly trending higher in KD than SD. It is unclear why this is not significant (i.e., if different n(mice) are causing non-statistical significance).

We now further validated this data by performing occludin staining and assessing occludin expression via qPCR in the colons of KD and SD fed mice. We were not able to identify a difference between the two diet-fed mice groups (Extended Data Fig.6e and 6f). These findings

are aligned with studies of KD in mice, in which the KD maintains a robust mucus layer despite the lack of fermentable carbohydrates and the KD-driven gut microbial shift is independent of colonic mucin¹⁰. Furthermore, no difference in LPS levels in plasma from KD-fed mice compared to SD-fed mice was observed (Extended Data Fig. 5c and 5d), indicating no detectable difference in bacterial translocation that could be further pursued.

Updated text in lines 149-152:

We first excluded bacterial translocation as a primary factor in the KD-mediated phenotype, as mucus layer thickness (Extended Data Fig. 5a and 5b (GF)), LPS levels in plasma (Extended Data Fig. 5c (GF) and 5d (SPF)), and occludin expression (Extended Data Fig. 5e and 5f (SPF)) remained unaltered upon KD consumption, in line with Ang et al.

New Extended Figure 5. e-f. Colonic occludin gene expression (**e**) and protein expression (**f**) in KD and SD mice in the SPF dietary experiment, as measured by qPCR (**e**) and immunofluorescent staining (occludin-TRITC in red and DAPI nuclear stain in blue, one representative colon per condition is shown, scale bar = 200 μ M, **f**). Two-tailed Mann-Whitney U test in (**a**), (**c**), (**d**), (**e**) and (**g**). Data in (**a**) shows n=4 SD and n=6 KD mice. Data in (**c**) and (**d**) shows n=4 mice per condition. Data in (**e**) shows n=6 mice per condition.

References

1. Tosolini, M. *et al.* Clinical impact of different classes of infiltrating T cytotoxic and helper cells (Th1, th2, treg, th17) in patients with colorectal cancer. *Cancer Res.* **71**, 1263–1271 (2011).

2. Blatner, N. R. *et al.* Expression of ROR γ t marks a pathogenic regulatory T cell subset in human colon cancer. *Sci. Transl. Med.* **4**, 164ra159 (2012).
3. Hyun, Y. S. *et al.* Role of IL-17A in the development of colitis-associated cancer. *Carcinogenesis* **33**, 931–936 (2012).
4. Parker, B. J., Wearsch, P. A., Veloo, A. C. M. & Rodriguez-Palacios, A. The Genus *Alistipes*: Gut Bacteria With Emerging Implications to Inflammation, Cancer, and Mental Health. *Front. Immunol.* **11**, 906 (2020).
5. Sohlenkamp, C. & Geiger, O. Bacterial membrane lipids: diversity in structures and pathways. *FEMS Microbiol. Rev.* **40**, 133–159 (2016).
6. Caswell, B. T. *et al.* Thioesterase enzyme families: Functions, structures, and mechanisms. *Protein Sci.* **31**, 652–676 (2022).
7. Han, S. *et al.* A metabolomics pipeline for mechanistic interrogation of the gut microbiome. *Nature* **595**, 415–420 (2021).
8. Grivennikov, S. I., Greten, F. R. & Karin, M. Immunity, inflammation, and cancer. *Cell* **140**, 883–899 (2010).
9. Ang, Q. Y. *et al.* Ketogenic Diets Alter the Gut Microbiome Resulting in Decreased Intestinal Th17 Cells. *Cell* **181**, 1263-1275.e16 (2020).
10. Wei, S.-J. *et al.* Ketogenic diet induces p53-dependent cellular senescence in multiple organs. *Sci. Adv.* **10**, eado1463 (2024).
11. Jukic, A., Bakiri, L., Wagner, E. F., Tilg, H. & Adolph, T. E. Calprotectin: from biomarker to biological function. *Gut* **70**, 1978–1988 (2021).
12. Ternes, D. *et al.* The gut microbial metabolite formate exacerbates colorectal cancer progression. *Nat. Metab.* **4**, 458–475 (2022).

13. Zhang, L. *et al.* The adhesin RadD enhances *Fusobacterium nucleatum* tumour colonization and colorectal carcinogenesis. *Nat. Microbiol.* **9**, 2292–2307 (2024).
14. Huang, P. *et al.* *Peptostreptococcus stomatis* promotes colonic tumorigenesis and receptor tyrosine kinase inhibitor resistance by activating ERBB2-MAPK. *Cell Host Microbe* **32**, 1365-1379.e10 (2024).
15. De Nies, L. *et al.* Evolution of the murine gut resistome following broad-spectrum antibiotic treatment. *Nat. Commun.* **13**, 2296 (2022).

We would like to thank the reviewers for their valuable comments. Please find our point-by-point response to their comments below.

>Major comments:> 1. & 2. The authors added experiments to investigate the effects of a stearic acid-rich diet on humanized gut microbiome-transplanted germ-free mice in an AOM/DSS model, finding a reduced tumor burden accompanied by a decrease in Th17 cells. KD diet-treated mice show slightly better suppression of tumor burden (Fig. 1c vs. New Fig. 6e.) Can the absolute concentration of stearate be measured and compared between KD diet-treat mice and stearic acid-rich diet-treat mice? Additionally, please add “GF” in New Figure 6a to indicate that germ-free mice were used in this figure. >

We have now added a deuterium labeled stearic acid (C18:0-d5, Eurisotop DLM-2712) as internal standard to enable absolute quantification of stearic acid (C18:0). The new data is shown in Fig 6d and Extended Fig. 7a. The results show that the stearic acid levels are similar in the KD and the stearic acid-fed mice.

New Extended Figure 7 a. Stearic acid levels detected in fecal samples from SPF KD- and SD-fed mice (Fig. 4c) by GC-MS (µg/mg of feces in non-polar phase).

New Figure 6c-d. Stearic acid concentrations detected in serum (µg/µL, c) and fecal (µg/mg, d) samples from CD- and SSD-fed mice, as detected by LC-MS.

We have updated the main text and highlighted the changes in lines 201 and 249 as shown below (highlighted in yellow) and in the manuscript.

Line 201: A single detected long-chain fatty acid (LCFA), namely free stearic acid (C18:0, also known as octadecanoic acid or stearate), was elevated in KD (Fig. 4c, Extended Data Fig. 7a **for absolute quantities**) and KC (Fig. 4d) mice.

Line 249: Mice on stearate-supplemented diet (SSD) also exhibited elevated stearic acid levels in serum and fecal samples (Fig. 6c and d, Supp. Tables 7 and 8), **similar to the levels observed in the fecal samples of mice on KD (Extended Data Fig. 7a)**.

Supplementary File 1 one contains information on standards used for absolute quantification.

Lines 256-258: The deuterium labelled tridecanoic acid (C13:0-d25, Eurisotop D-4002) was used as internal standards to enable absolute quantification of stearic acid (C18:0) in the non-polar phase of fecal samples analyzed by GC-MS (linked to Extended Fig. 7a).

Lines 343-345: The deuterium labelled stearic acid (C18:0-d5, Eurisotop DLM-2712, 2 µg/ml in the extraction fluid) was used as internal standards to enable absolute quantification of stearic acid (C18:0) (linked to Figure 6c-d).

> 4. In the authors' response "In our study, KD-fed mice displayed decreased Th17 cells and associated cytokines, suggesting that the ketogenic diet reduces this immune cell subtype...". However, the results showing reduced cytokines are not presented in the paper. Please clarify this point.

Thanks for raising this point. The sentence referred to IL17 as we used IL17 staining in the flow cytometry to identify IL17 producing T cells, thus the sentence in the rebuttal letter should read "In our study, KD-fed mice displayed decreased IL17 producing T cells, suggesting that the ketogenic diet suppresses this immune cell subtype".

Updated lines in the manuscript 101-104 in which we now clarified that we refer to IL17 producing T cells.

We particularly observed a decrease in IL17 producing T cells (CD4+ IL17+, Extended Data Fig. 2f), a pathogenic subpopulation of CD4+ T cells known to play a critical role in CRC development as evidenced by reduced expression of IL-6, STAT3, IFN γ , and TNF α , leading to fewer and smaller tumors in IL-17A-deficient mice.

Associated figure legend in lines 1034 (already within the first version of the revised manuscript) f. CD4+ IL-17+ T cells (counts) in the colons of SD and KD mice.

> 6. “We observed a reduction in IL17 levels, consistent with our previous in vivo findings in KD-fed mice.” These data of IL17 levels in KD-fed mice could not be found

Thanks for pointing this out. We want to clarify that we observed a reduction in colonic mRNA IL17 levels, consistent with our previous observation of reduced numbers of IL17-producing T cells in the lamina propria of KD-fed mice.

We had initially referred to the lines in the rebuttal letter but indeed had omitted to add the lines in the updated manuscript. We apologize for this oversight. Please find them here below.

Updated text in lines 159-161:

Furthermore, colonic mRNA levels of IL17 were decreased in KC mice compared to SC mice (Extended Data Fig. 6e), in line with the decreased number of IL17 producing T cells in the KD-fed mice (Extended Data Fig. 2f).

Figure legend (already present within the first revised manuscript): line 1090 further clarifies that we are referring to mRNA levels. d-e. S100A8 (e) and IL-17 (e) gene expression in KC and SC mice in the CMT experiment, as measured by qPCR. Data is shown as mean \pm SD in (a), (b), (d) and (e).

Minor comments. Typographical issue: Please correct “C57BL6/J” and “C57BL6/NTac” (eg. Fig.1 a and Fig. 2a) to C57BL/6J and C57BL/6NTac.

Thanks for pointing this out, we have corrected the figures.

We look forward to reading from you.

Sincerely yours,